# MAP INDUCTION: COMPOSITIONAL SPATIAL SUBMAP LEARNING FOR EFFICIENT EXPLORATION IN NOVEL ENVIRONMENTS

**Sugandha Sharma, Aidan Curtis, Marta Kryven, Josh Tenenbaum, Ila R. Fiete**
Massachusetts Institute of Technology
`{susharma,curtisa,mkryven,jbt,fiete}@mit.edu`

## ABSTRACT

Humans are expert explorers and foragers. Understanding the computational cognitive mechanisms that support this capability can advance the study of the human mind and enable more efficient exploration algorithms. We hypothesize that humans explore new environments by inferring the structure of unobserved spaces through re-use of spatial information collected from previously explored spaces. Taking inspiration from the neuroscience of repeating map fragments and ideas about program induction, we present a novel "Map Induction" framework, which involves the generation of novel map proposals for unseen environments based on compositions of already-seen spaces in a Hierarchical Bayesian framework. The model thus explicitly reasons about unseen spaces through a distribution of strong spatial priors. We introduce a new behavioral Map Induction Task (MIT) that involves foraging for rewards to compare human performance with state-of-the-art existing models and Map Induction. We show that Map Induction better predicts human behavior than the non-inductive baselines. We also show that Map Induction, when used to augment state-of-the-art approximate planning algorithms, improves their performance.

## 1 INTRODUCTION

Humans efficiently use spatial reasoning to explore and forage in new environments. We easily find our way around new buildings and infer promising foraging locations. For instance, after some experience of foraging on a cluster of hills and finding berries on the south-facing slopes, we could anticipate that nearby hills may have a similar distribution of berries (see Figure 1).

Which neurally-informed computational cognitive mechanisms make this possible, and could they be leveraged for better exploration in AI? We empirically study human exploration in novel spaces, and propose a formal computational model, which we call Map Induction, that predicts human behaviour by leveraging Bayesian inference about the structure of unobserved space. We also show that map induction can improve the performance of a Partially Observable Markov Decision Process (POMDP) planner during foraging. We begin by reviewing the literature on human spatial cognition involved in exploration and parallel developments of exploration algorithms, followed by introducing our computational modeling and experimental results.

**Distributed non-metric representations.** Human exploration relies on constructing approximate spatial representations (Wang & Brockmole, 2003; Warren et al., 2017; Wiener & Mallot, 2003) that support near-optimal planning in daily life (Bongiorno et al., 2021) while sacrificing global metric accuracy (Vuong et al., 2019; Newcombe et al., 1999; Zhu & Levinson, 2015). These representations are acquired by combining redundant observations of local landmarks, such as views from multiple perspectives (Gillner & Mallot, 1998; Foo et al., 2005). The use of distributed spatial representations

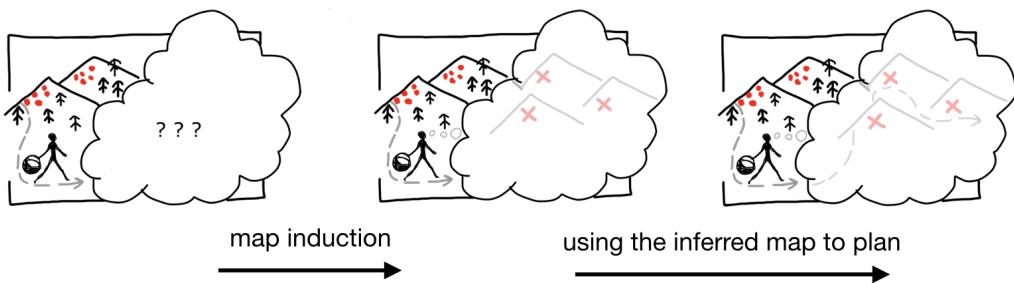

Figure 1: Map induction: We propose how humans might infer the structure of unobserved spaces based on priors constructed from observations of other environments or different parts of the same spacee. Map induction can minimize exploration cost by filling in incomplete or missing observations from experience.

for cost-effective navigation also has a long history in AI. Examples include a computational theory of wayfinding based on multiple locally unique but globally similar landmarks (Prescott, 1996), which prioritized error recovery over accuracy. The Spatial Semantic Hierarchy model (Kuipers & Levitt, 1988) used distributed spatial graphs as maps for navigation (Kuipers, 2000; Kuipers et al., 2004). Simultaneous Localization and Mapping (SLAM) algorithms based on segmenting the explored space circumvent scale limitations by dividing the world into manageable locally metric submaps with topological relationships between submaps (Bosse et al., 2003; Fairfield et al., 2010). By contrast, recent recurrent neural network models of navigation have focused more closely on the metric structure of explored spaces and global map formation (Burak & Fiete, 2009), showing how networks trained on path integration can take shortcuts (Banino et al., 2018).

**Hierarchical organization.** Humans minimize planning costs by representing spaces hierarchically – based on visible spatial boundaries (Kosslyn et al., 1974), geography (Stevens & Coupe, 1978), or otherwise interpreting spaces as composed of sub-regions (Hirtle & Jonides, 1985). Hierarchical spatial organisation is evident in a tendency to plan paths between regions, before planning sub-paths within each region (Bailenson et al., 2000; Newcombe et al., 1999; Wiener & Mallot, 2003; Wang & Brockmole, 2003; Balaguer et al., 2016), and in increased reaction times when switching between hierarchy levels during plan execution (Balaguer et al., 2016; Kryven et al., 2021). Hierarchical state-spaces in non-spatial planning domains include drawing (Tian et al., 2020) inverse inference (Shu et al., 2020), and reasoning about topology (Tomov et al., 2020).

Hierarchical representations in AI and specifically in Reinforcement Learning can be expressed as options – a hierarchical grouping of actions frequently performed together (Sutton et al., 1999). Assuming a novel Markov Decision Processes (MDP) is sampled from a family of similar MDPs, its rewards can be learned as derived from a parent distribution shared by the MDP family (Wilson et al., 2012). Singh et al. (2012) proposed a computational framework for learning efficient state-space representations by recognizing which sequences of actions lead to identical observations – for example, grouping together paths that lead to observing the same landmark. A principle of grouping game-board states based on rotation and reflection symmetries has been used to optimize state-space representations in the game of Go (Silver et al., 2017).

**Shared reference frames.** From young children to hunter-gatherers, humans spontaneously organize sensory precepts into patterns (Pitt et al., 2021), and use them to generalize between tasks (Tian et al., 2020; Lake & Piantadosi, 2020; Schulz et al., 2018). Generalization and transfer learning in spatial domains include mirror-invariant neural scene representations (Dilks et al., 2011), and reusing of reference frames across similar environments (Marchette et al., 2014). Shared reference frames may occur in other mammals as well, as suggested by evidence of rodents reusing grid-cell maps between similar parts of an environments (Derdikman et al., 2009; Carpenter et al., 2015). A recent computational framework for clustering space by fragmenting neural representations at locations of high surprise during online exploration shows how such submaps might form (Klukas et al., 2021). However, these studies have not considered how submaps may be recombined and reused to efficiently forage in novel spaces.

**Map Induction.** We propose a *Map Induction* hypothesis – that humans optimize exploration of new spaces by representing maps as composed of reusable reference frames – which can inferred

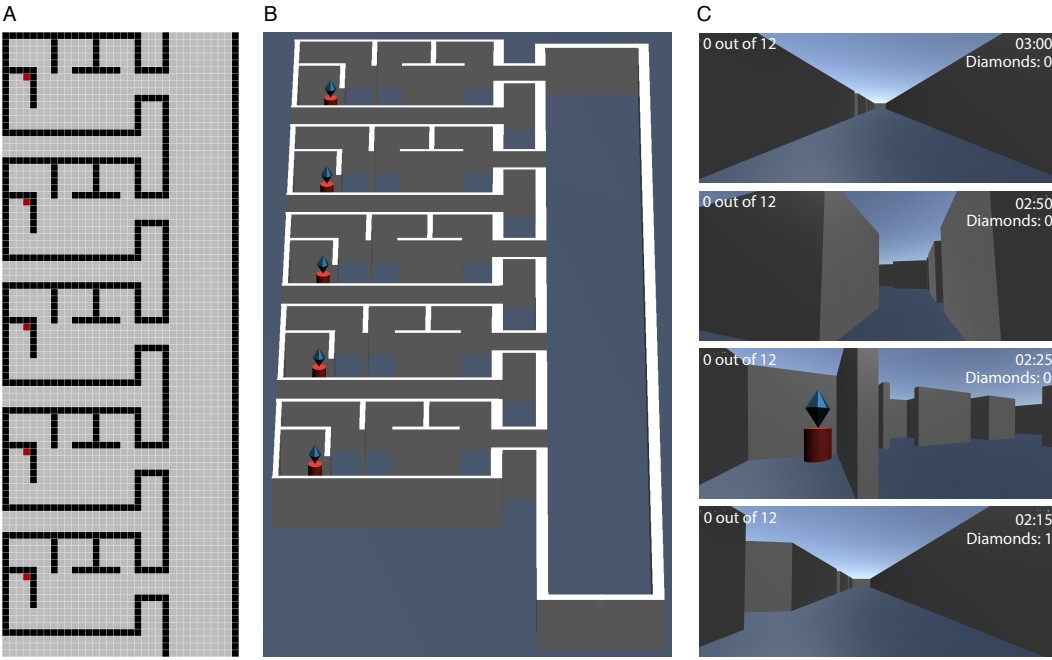

Figure 2: An example of an MIT task. **A.** A 2-dimensional grid discretization showing the map used for modeling exploration trajectory. The black cells indicate walls, and the red cells indicate reward locations. **B.** The 3-dimensional world, in which subjects search for rewards (shown as blue diamonds on a red platform), seen from an overhead perspective in the Unity Design system. **C.** First-person views of the task, as seen by subjects at different times during the experiment.

by program induction to represent unseen space as composed of previously encountered regions. *Program induction* generally refers to inferring a *program* that likely produced a given sequence of observations (Stuhlmuller et al., 2010; Lake et al., 2015). For example, given a sequence of numbers, the inferred program may be an arithmetic progression; given a natural texture the program may be a reaction-diffusion equation (Camazine et al., 2020). Although Bayesian models of concept-learning by program induction have been developed for many domains (Stuhlmuller et al., 2010; Lake et al., 2015; Tian et al., 2020), its use to understand spatial structures and environments is unexplored. In this work *map induction* refers to inferring program(s) that could generate the current environment, given past observations. Given such a program, an agent can anticipate the structure of an unseen environment and use this estimated structure to optimize exploration.

In this work we adopt a scientific and an engineering goal: (1) to empirically study how humans learn spatial representations, and (2) describe a computational model that formalizes human-like map induction that can potentially optimize exploration in AI. We show that a Partially Observed Markov Decision Process (POMDP) planner augmented with Hierarchical Bayesian map induction predicts human exploration better than a naive, non-inductive, POMDP (Experiment 1). We also show that human exploration relies on probabilistic distributions over likely map-generating programs, as opposed to using only the most likely map (Experiment 2). In the next section, we introduce the Map Induction Task. We then give a detailed description of computational models in Section 3. In Section 4, we describe two human experiments, and compare behavioral metrics with our models' predictions.

## 2 MAP INDUCTION TASK

The Map Induction Task (MIT) is designed to present subjects with novel environments and provide a context for learning novel spatial representations. We are especially interested in environments in which certain parts of the environment can be predicted from incomplete observations. For example, upon entering a unit in an apartment building a human will likely categorize it as a new region, but after exploring several units, one should be able to anticipate the remaining floor-plan, or even

consider several possibilities consistent with prior observations – such as, that the right and the left wing of the building could mirror each other. In nature, predictable structures occur in patterns of hills and valleys, branching riverbeds, and plants favoring certain features of the environment (see Figure 1), which in theory can be probabilistic, rather than exactly structured repetitions.

The subjects' task is to forage for rewards in novel partially observable 3-dimensional environments, which contain multiple rewards. To keep each experiment duration under 45 minutes, in the current experiments each environment consisted of repeated *units* – structural blocks repeated throughout the environment (see Figure 2B). However, in theory the environment structure could be a probabilistic mixture of different parts.

The subjects have no advance knowledge of the specific environment structure and are instructed to collect all diamonds (see Figure 2 C). Subjects use a keyboard and mouse (or a trackpad) to navigate, which are standard navigation controls in first-person games. The distribution of rewards is initially unknown to the subjects, but is predictable from the layout of the environment – for example, the rewards may be placed in the same part of a unit each time a the unit occurs. While the MIT task can be completed without map induction, or even without an explicit map representation (e.g. by following a wall) such a naive strategy would take longer than selectively exploring only the parts likely to contain diamonds. Each trial continues either until all diamonds are collected or until a timeout is reached, whichever comes first.

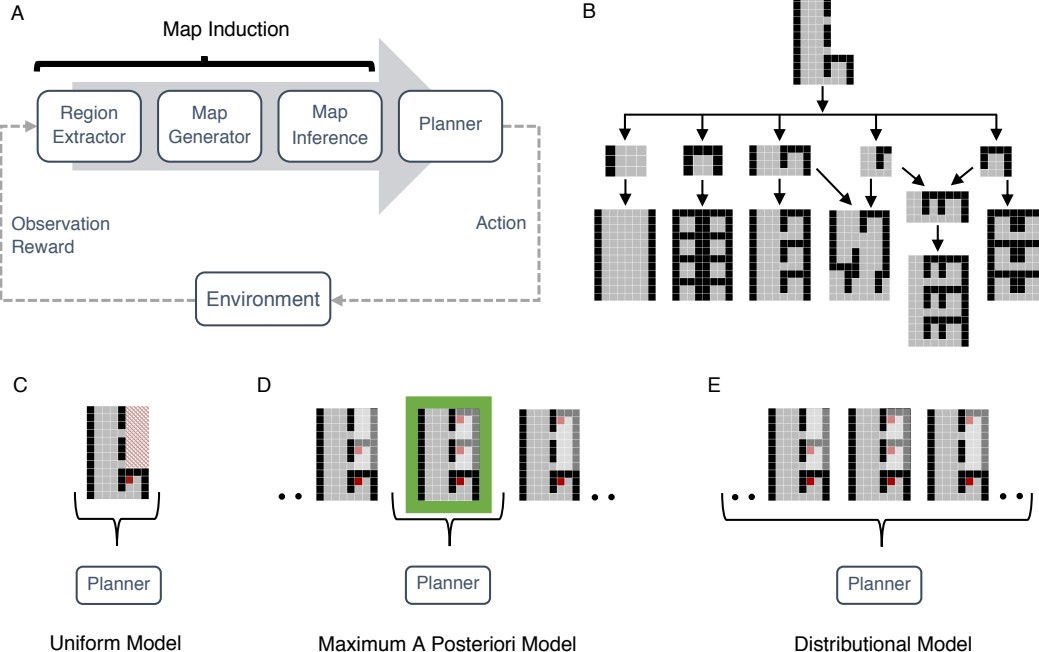

Figure 3: Model architecture. **A.** Four steps required to solve the MIT task and the corresponding computational modules in our framework. **B.** Extracting the candidate regions (submaps) from the known map, and generating possible map completions by composing these regions and their simple transformations (e.g., mirror reflections) as defined in the generative grammar of the Map Generator. **C.** Uniform Model assumes a uniform distribution for the unobserved part of the map. **D.** Maximum A Posteriori Model uses the most likely map completion for planning. **E.** Distributional Model uses the entire distribution of map completions for planning.

## 3   COMPUTATIONAL MODELS

We now formalize the map induction hypothesis to quantitatively and computationally test its predictions. Optimizing performance in the MIT task requires the following computational steps: (1) inferring that the environment is composed of repeating units (2) inferring the most likely structure(s) of the environment (3) inferring the distribution of rewards within the environment, and (4)

planning the shortest route that collects all rewards. Here (1) and (2) refer to map induction, and (3) may involve further program induction over the distribution of rewards (e.g. the rewards could be placed in the top left corner of every unit, in every odd unit, etc.). Because of uncertainty in the environment, this computational pipeline is executed in the context of a replanning loop. Finally, step (4) involves solving a Partially Observed Markov Decision Process (POMDP) that uses a distribution of hypothetical environments to plan. While it is possible to do (4) without performing the first three steps, this would entail assuming an uninformed prior that any unobserved cell could equally likely be a wall, an empty space, or a reward – which would result in exploring exhaustively.

Formally, we can model this process by a Bayesian generative framework consisting of four computational modules (see Figure 3A) corresponding to the four steps described above. Each module can be independent of the others and can express different modeling assumptions. In the remainder of this section, we discuss the function of each module in detail and define three specific combinations of module implementations, to express three distinct computational hypotheses formalizing the principle of map induction in different ways.

## 3.1 MAP INDUCTION THROUGH OBSERVED SPATIAL STRUCTURES

Steps (1) and (2) entail approximate inference of a posterior distribution $p(M|\mathcal{D})$ over possible maps. Here we formalize the computations that estimate this distribution (see Figure 3A).

*Region Extraction:* The region extractor extracts candidate regions (or submaps) $M_p$ from known parts of the map (see Figure 3B for a simple example). The second stage of the hierarchy in Figure 3B shows examples of regions extracted from the observed map, shown above. Note that the term *region* refers to a hypothetical building block of the environment considered by the model, and is different from the term *unit* used in the previous section to describe repeating elements in environment design. In theory multiple regions can represent a unit, or a region may comprise several units.

*Map Generation:* To develop a space of possible map completions, we use a probabilistic generative grammar that combines region primitives into a set of completed maps. If the current map has dimensions $s_x, s_y$, and $d_x, d_y$ are the dimensions of the extracted regions, then the generative grammar for that map is described as shown in Table 1. The third level of the hierarchy in Figure

| Production Rule | | | Probability |
|---|---|---|---|
| $M_p(d_x, d_y)$ | $\rightarrow$ | FLIPH$(M_p(d_x, d_y))$ | $\frac{1}{3}$ |
| $M_p(d_x, d_y)$ | $\rightarrow$ | FLIPV$(M_p(d_x, d_y))$ | $\frac{1}{3}$ |
| $M_p(d_x, d_y)$ | $\rightarrow$ | ROTATE90$(M_p(d_y, d_x))$ | $\frac{1}{3}$ |
| $M_p(d_{x_1} + d_{x_2}, d_{y_1})$ | $\rightarrow$ | HCAT$(M_p(d_{x_1}, d_{y_1}), M_p(d_{x_2}, d_{y_1}))$ | $\frac{1}{2}$ |
| $M_p(d_{x_1}, d_{y_1} + d_{y_2})$ | $\rightarrow$ | VCAT$(M_p(d_{x_1}, d_{y_1}), M_p(d_{x_1}, d_{y_2}))$ | $\frac{1}{2}$ |
| $M$ | $\rightarrow$ | $M_p(s_x, s_y)$ | $1$ |

Table 1: Probabilistic context-free grammar used to generate a distribution of map hypotheses.

3B shows examples of map completions generated this way. The probabilities assigned to the rules in the generative grammar encode priors on coverage and consistency of the generated maps.

*Map Inference:* We use this probabilistic generative grammar along with the following likelihood over generated maps to form the posterior $p(M|D)$:

$$l(M; D) \propto \left[1 - \frac{\beta}{s_x \times s_y}\right] \times \left[1 - \frac{\gamma}{s_x \times s_y}\right]$$

where $\gamma/s_x \times s_y$ is the fraction of the map not predicted by a given map completion and $\beta/s_x \times s_y$ is the fraction of mismatched reward locations relative to the history of observations. At a high level, this likelihood function encodes two simple assumptions: (1) map completions that are maximally

descriptive of unseen portions of the environment are more likely; (2) map hypotheses that are minimally contradictory with previous observations are more likely. The posterior distribution $p(M|D)$ is computed using the above likelihood function, and an initial prior on the map hypotheses $p(M)$ derived from the probabilities assigned to the production rules in the generative grammar listed in Table 1.[1] For simplicity, here we assume a uniform prior over the map hypotheses generated by the Map Generator since the space of possible hypotheses is small.

$$p(M|D) = l(M; D)p(M) = p(D|M)p(M)$$

### 3.1.1 PLANNER

We model the exploration problem as a Partially Observable Markov Decision Process (POMDP). In a POMDP, the state is not directly observable by the agent and can only be inferred through sequences of observations in the environment. A POMDP can be described as a tuple $\langle \mathcal{S}, \mathcal{A}, T, R, \Omega, \mathcal{O}, \gamma \rangle$ where $\mathcal{S}$ is the set of possible states in the world, $\mathcal{A}$ is a finite set of actions, $T : \mathcal{S} \times \mathcal{A} \to \Pi(\mathcal{S})$ is a stochastic transition function which maps a state action pair to a distribution over possible next states, $R : \mathcal{S} \times \mathcal{A} \to \mathbb{R}$ is a reward function that maps state action pairs to a scalar reward, $\Omega$ is a set of observations that an agent can experience in the world, $\mathcal{O} : \mathcal{S} \times \mathcal{A} \to \Pi(\Omega)$ is an observation function which maps a state and action pair to a distribution of possible observations after taking the action, and $\gamma$ is a discount factor such that $0 < \gamma < 1$. An optimal agent in this formulation acts to maximize the expected discounted reward from the environment.

$$\pi^*(s) = \arg\max_a \left( \mathbb{E}\Big[ \sum_{k=0}^{\infty} \gamma^k R(s, a) \Big] \right)$$

For our task, the state space is defined as $\mathcal{S} = \mathcal{H} \times [0, 1]^{|\mathcal{H}|} \times \mathcal{S}_a$ where $\mathcal{H}$ is the set of possible map hypotheses from $p(M|D)$, $[0, 1]^{|\mathcal{H}|}$ is the probability associated with each hypothesis, and $\mathcal{S}_a$ is a 3-tuple $(a_x, a_y, a_r)$ with dimensions $(s_x, s_y, 4)$ indicating the index into the grid world state where the agent is positioned and the current direction the agent is facing. The action space is discrete with size 3 for bidirectional rotation and forward movement. We use a reward function $R$ defined by an indicator function over the state space $\mathbb{1}(S_g[a_x, a_y] = \text{reward})$. The transition function $T(s'|s, a)$ is a stochastic function that maps the given action and the previous state to a new state. There is no closed-form expression for $T$, but it can be expressed using a generative probabilistic sampler that takes in $s$, samples a grid from the hypothesis distribution in $s$, deterministically simulates action $a$ in the sampled grid, and returns a new position, orientation of the agent and an updated hypothesis distribution consistent with the new observations. The observation space $\Omega(s, a; D)$ is defined by a surjective line-of-sight (LOS) observation function $\mathcal{O}$ that maps from state and action to a deterministic observation. This function casts rays within a 90-degree field of view in the direction of the agent's orientation. Lastly, we define $\gamma = 0.90$ for all of our experiments. Since finding an exact solution is intractable due to the size of the problem (Kaelbling et al., 1998), we search through belief space using an approximate online Partially Observable Monte Carlo Planner (POMCP) (Silver & Veness, 2010).

### 3.2 COMPUTATIONAL HYPOTHESES

We compare human performance to three hypotheses (variants of POMCP) to evaluate whether and how human exploration implements the computational steps described in Figure 3A:

1. **Uniform Model (Uniform-POMCP)**: The Uniform model doesn't use map induction, i.e., it doesn't learn any inductive priors about spatial structures during exploration outlined in steps (1)-(3). Instead, it assumes a uniform distribution over possible settings for each grid cell (reward, empty, etc.) and uses that for planning as shown in Figure 3C.

2. **Maximum A Posteriori Model (MAP-POMCP)**: The MAP model uses the most likely map from the distribution of induced maps $p(M|D)$ to plan (see Figure 3D).

---

[1]Given the relatively small environments, we are able to compute the posterior exactly. However, larger and more complex environments (where enumerating all possible map completions is not feasible) would likely require approximate Bayesian inference methods such as Markov Chain Monte Carlo.

3. **Full Distributional Model (D-POMPC)**: The Distributional model uses the entire set of induced maps in the distribution $p(M|D)$ to plan (see Figure 3E).

Both second and third hypotheses are consistent with the previous studies discussed in Section 1. They both use the map induction framework based on compositionality of space. i.e., combining local regions to form the global map instead of learning a global metric representation.. On the other hand, the first hypothesis is inconsistent with this literature, since it makes no prior assumptions about the organization of space (neither hierarchical organization, nor the use of shared reference frames). Furthermore, it assumes a uniform distribution over the unseen map space which is akin to learning a global map.

## 4 BEHAVIORAL EXPERIMENTS

Below we describe two human behavioral experiments. Experiment 1 tests whether humans perform map induction, as implemented by the MAP-POMCP and D-POMCP models, in contrast to naive exploration implemented by Uniform-POMCP. The results show that humans indeed rely on map induction and rule out the use of the Uniform model. Experiment 2 tests whether human exploration is best explained by MAP-POMCP or D-POMCP models, and shows that humans plan according to a distribution of hypothetical maps, explicitly gathering information to disambiguate those hypotheses. The number of subjects recruited for both experiments was determined through prior pilot experiments, our goal was to recruit a sufficient number of responses to differentiate between the Uniform-POMCP, MAP-POMCP and D-POMCP.

### 4.1 EXPERIMENT 1: USING MAP INDUCTION TO OPTIMIZE EXPLORATION

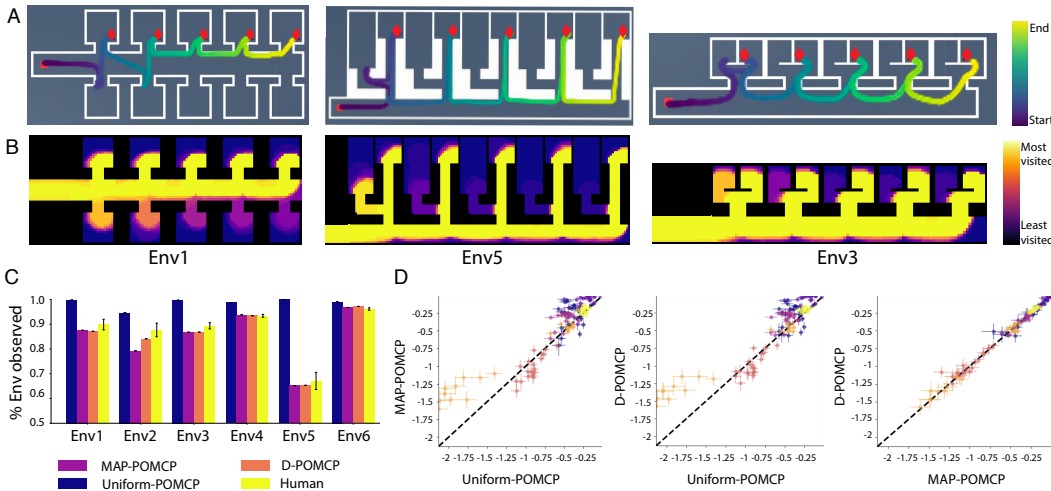

Figure 4: Experiment 1 results. **A**. Exploration trajectories of a representative subject. **B**. Visitation heatmaps of three example environments aggregated across subjects and pairs of reflected environments. The visited area was computed in a 2D grid projection, using a circular radius of five grid cells around the agent. **C**. Fractions of environments observed by the models and humans. Error bars show 95% confidence intervals. **D**. Log-likelihood that the model takes the same actions as the human (details in Appendix section A.3). Each marker is a subject-environment pair, with each color showing a single environment: Env1-Env6 (blue-yellow). Error bars show standard error along each axis.

*Method*: The experiment was conducted in a web browser, using 3D virtual environments built-in Unity WebGL. Each environment was composed of a corridor connecting five instances of a specific unit. The rewards (diamonds) were placed at identical locations within each unit. An example environment is shown in Figure 2. Each subject performed four practice trials, followed by an instruction quiz and 12 test trials in order randomized between subjects. Subjects who failed the instruction quiz repeated the instructions and practice until they answered the quiz correctly. The

timer for the trial and the number of diamonds collected were shown on the upper right corner of the screen (see Figure 2C). See Supplementary Materials for full experimental instructions and screenshots.

*Stimuli*: The 12 test trials comprised six pairs of unique environments and their reflections. The reflected environments increase the number of trials and control for possible left or right biases.

*Subjects*: We recruited 30 subjects via Amazon Mechanical Turk. Subjects were paid for 45 minutes of work and received a monetary bonus for each collected diamond.

*Results*: Out of the 30 subjects, 4 subjects explored exhaustively, 2 did not complete the experiment, and 24 subjects correctly inferred reward placement, indicative of map induction. Most subjects exhaustively explored the first two units in a new environment, followed by partial exploration through the remainder of the trial. Throughout the experiment, the extent to which non-rewarding areas were visited decreased, indicating that people were increasingly confident about the repeating layout.

Figure 4A shows examples of exploration trajectories of one subject. Subjects generally used identical trajectories to search the reflected pairs of environments, and so we aggregated results for each of the six pairs. Figure 4B illustrates the extent to which three example environments were visited, aggregated across subjects, and across reflected pairs of environments. See the Appendix for a complete plot of all exploration trajectories and heatmaps. Figure 4C shows the fractions of each environment observed by humans and models. The areas observed by MAP-POMCP and D-POMCP were similar to humans', while the area observed by Uniform-POMCP was significantly larger. Figure 4D shows the fit of the three models to human behavior, indicating that the MAP-POMCP and D-POMCP models, which implement map induction, explain human behavior better than the Uniform-POMCP model, which makes no predictions about map structure.

## 4.2 EXPERIMENT 2: DISTRIBUTIONAL SUB-MAP REPRESENTATIONS

Experiment 2 differentiates between MAP-POMCP and D-POMCP models using color cues to indicate the location of rewards within each unit. This was done by including a 'cue room' within each unit – a small room containing information about the location of the reward. The environments were designed so that MAP-POMCP would take a longer exploration path compared to D-POMCP. A planner guided by the MAP-POMCP model ignores the cue rooms and heads toward the part of a unit most likely to contain a diamond based on previous experience. In contrast, the D-POMCP model visits the cue room to gather information about the location of the diamonds.

*Method*: Experiment 2 followed a procedure similar to Experiment 1, with a different set of environments. Each subject performed five practice trials, followed by an instruction quiz, 12 test trials in order randomized between subjects, and a test of skills using navigation controls. The controls skill test ensured that subjects could navigate in WebUnity without undue difficulty since Experiment 2 used larger environments. Subjects who failed the controls skill test were paid but excluded from the analysis. The instructions were modified to prompt subjects to consider colors as cues to the placement of the reward – see the Appendix for full details.

*Stimuli*: The 12 test environments comprised six unique environments and their reflections. The color cues were randomized within each environment so that the reflected pairs of environments were colored in different ways and contained rewards in different locations.

*Subjects*: We recruited 51 subjects via Amazon Mechanical Turk. Each was paid for 45 minutes of work and received a monetary bonus for each collected diamond.

*Results*: Out of 51 subjects, 35 completed the controls skill test. Of the 35 subjects, all but nine were able to successfully use color cues, as predicted by the D-POMCP model, indicating that the majority took into account the entire distribution of possible maps in contrast to just the most likely map. Example exploration trajectories of one subject and visitation heatmaps are shown in Figure 5 A and B. See the Appendix for more results. Comparing the fractions of environments observed by humans and models indicates that only the D-POMCP model was comparable to humans (see Figure 5C). D-POMCP model was also best at explaining human behavioral data, as shown in Figure 5D.

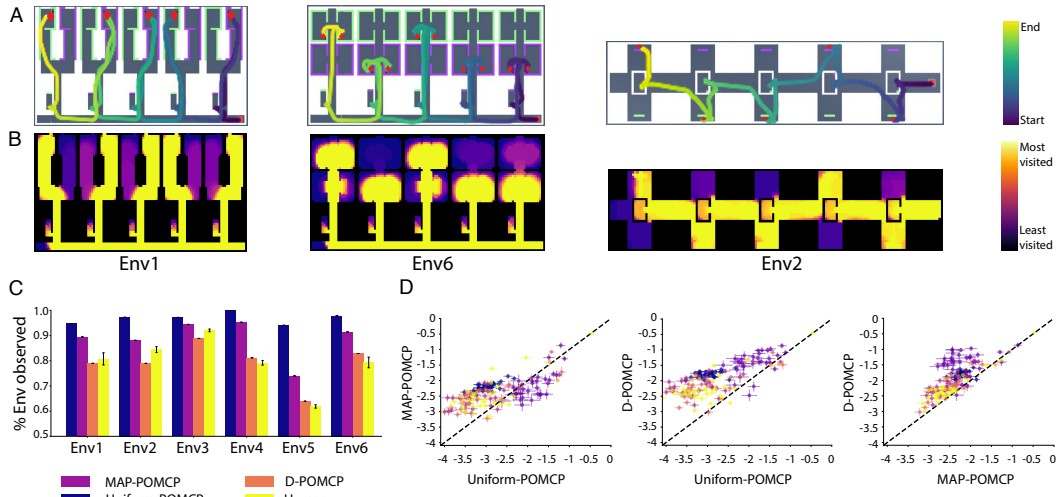

Figure 5: Experiment 2 results. **A**. Exploration trajectories of a representative subject. **B**. Visitation heatmaps of three example environments aggregated across subjects. The visited area was computed in a 2D grid projection, using a circular radius of five grid cells around the agent. **C**. Fractions of environments observed by the models and humans. Error bars show 95% confidence intervals. **D**. Log-likelihood that the model takes the same actions as the human (details in Appendix section A.3). Each marker is a subject-environment pair, with each color showing a single environment: Env1-Env6 (blue-yellow). Error bars show standard error along each axis.

## 5 DISCUSSION

In this work we proposed a novel *Map Induction hypothesis* – that humans use program induction to infer maps of novel environments from partial observations and use the inferred map distribution to optimize exploration for rewards. We formalize this hypothesis computationally, by combining a Bayesian map induction model and an approximate belief-space planner. We present the results of two behavioral experiments that support the map induction hypothesis by demonstrating that our computational model predicts human exploration, and show that the performance of a Partially Observable Monte Carlo Planner can be improved by adding map induction.

While we explored map induction in simple environments, it is likely to apply more widely – humans not only forage selectively but also tend to consider only plausible theories. This may indicate that humans anticipate the structure of abstract search spaces by noticing repetitions and symmetries to simplify hard computing problems in various domains. Using map induction for exploration may not be unique to humans – the evidence of rodents reusing grid-cell maps in similar parts of environments suggests that grid-cell remapping may be a neural signature of map induction (Derdikman et al., 2009; Carpenter et al., 2015). Hippocampal reuse of place-cell maps in composite environments (Paz-Villagrán et al., 2004), which are invariant to rotation and scaling (Muller & Kubie, 1987), suggests that animals use landmark cues to guide map induction as well.

In future work we intend to study map induction in larger, more naturalistic environments, where more comprehensive generative models may be needed for map induction in order to optimally induce the generative programs used to generate these environments. Map induction may also have potential applications for fast and generalizable map learning in SLAM and in model-based RL tasks.

## ACKNOWLEDGMENTS

We would like to thank Kevin Ellis for helpful discussions, and Kevin M. Zayas for assistance with the development of web-hosting our tasks. This work is supported by the Center for Brains, Minds and Machines (CBMM), funded by NSF STC award CCF-1231216. We acknowledge support from the K. Lisa Yang Integrative Computational Neuroscience Center (ICoN) and the Friends of Mc-Govern Fellowship at the MIT McGovern Institute of Brain Research to Sugandha Sharma; from

the NSF Graduate Research Fellowship Program to Aidan Curtis; from the ONR, the Simons Foundation, and HHMI through the Faculty Scholars Foundation to Ila Fiete.

REPRODUCIBILITY STATEMENT

The source code for the models and the behavioral datasets used for this project are made available at the following GitHub repository, along with instructions on how to use them.
`https://github.com/s72sue/Map-Induction`

The compiled Unity WebGL build packages for both experiments presented in this paper are also provided at the above repository. The builds can be hosted using any web hosting service (e.g., simmer.io) in order to re-run the experiments and reproduce experimental results. The Unity source code used to generate these builds is available on request.

To ensure reproducibility, the behavioral experiments are also explained in detail in the Appendix.

Our institutional IRB approved all experiments.

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

## A APPENDIX

### A.1 EXPERIMENT 1

Figure 6 shows the stimuli used in the experiment, with practice stimuli in panel A and test stimuli in panel B. Each of the test stimuli is composed of a corridor connecting five instances of a specific unit. The rewards (diamonds) are placed at identical locations within each unit. The task was to navigate the environments and collect all the diamonds in the least amount of time. At the beginning of each trial, the subject is spawned at a fixed starting location (marked by a red floor-mat) in all environments. The subjects can then use the keyboard and mouse controls to explore the environment and find the diamonds. Note that there are six base test stimuli with the other six as their reflected versions. We use reflected versions to counter-balance the left and right turns required across environments during the experiment and to increase the number of experimental trials while saving effort on environment design. This experimental design can also be used to study how map induction generalizes to transformations of environments (these results are out of the scope of this paper and are therefore not presented). For instance, we noticed that subjects were able to use program induction across environments when they shared similar spatial structures.

Figure 7 shows the scenes displayed during the progression of the experiment. Figure 7A is the welcome scene that subjects see at the beginning of the experiment. Here, we confirm that the subjects consent to participating in the experiment voluntarily, and collect their age and gender information. Figure 7B is the next scene that provides instructions to the subjects about the task and the keyboard/mouse controls that they can use to navigate during the task. Figure 7C shows the first practice trial and is followed by three additional practice trials in the order shown in Figure 6A. At the end of each trial, a "Level Complete" (Figure 7D) or "Out of Time" (Figure 7E) pop-up window is displayed depending on whether the subject collected all the diamonds in the environment or ran out of time before doing so. Both these pop-ups also specify the ratio of the number of diamonds collected to the total diamonds embedded in the environment. This provides feedback to the subjects about their performance in the trial before they continue to the next trial. At the end of the practice trials, subjects are presented with an instruction quiz as shown in Figure 7F, to ensure that they understand the task. If they answer the instruction quiz incorrectly, they are requested to re-read the instructions and re-do the practice trials, and are sent back to the instructions scene in Figure 7B.

If the subjects answer the instruction quiz correctly, they are presented with the twelve test trials (see Figure 6B) in succession. An example test trial is shown in Figure 7G. The test stimuli are grouped into three blocks containing four stimuli each, according to their difficulty level. For each subject, the order of stimulus presentation within blocks is randomized, while the blocks themselves are presented in a fixed order e.g., block1 (simple environments - Env1, Env1R, Env2R, Env2R) is presented first, followed by block2 (moderate complexity - Env3, Env3R, Env4, Env4R), followed by block3 (high complexity - Env5, Env5R, Env6, Env6R). After the test stimuli, a final scene in Figure 7H is shown to thank the subjects for their participation, to get their comments about the strategies they used to solve the task, and their general feedback about the experiment.

Figure 8 shows the results from the experiment. Panel A shows the trajectories of a representative subject on six base test trials (Env1-Env6), shown in the order presented to the subject (top - bottom). Panel B shows the visitation heat-maps indicating the least to most visited regions in each environment, averaged across subjects. The heat-maps are aggregated across the base environments (Env1-Env3) and their reflected versions (Env1R-Env6R) shown in Figure 6B. Separate heat-maps for the base environments and their reflected versions are also shown in Figure 13 for comparison between the first and the second presentation (mirrored with respect to the first presentation).

## A.2   EXPERIMENT 2

Figure 9 shows the stimuli used in this experiment, with practice stimuli in panel A and test stimuli in panel B. The practice stimuli have color cues at the entrance indicating the location of diamonds. Each of the test stimuli is composed of a corridor connecting five instances of a specific unit. Unlike Experiment 1, the diamonds are placed at different locations within each unit determined by the color cue in the cue room of the unit. Relative to the base test environments (Env1-Env6), the reflected versions (Env1R - Env6R) have a reflected geometry, however unlike Experiment 1, they have a different reward distribution. The task was to navigate the environments and collect all the diamonds in the least amount of time. At the beginning of each trial, the subject is spawned at a fixed starting location (marked by a red floor-mat) in all environments. The subjects can then use the keyboard controls to explore the environment and find the diamonds.

Figure 10 shows the scenes displayed during the progression of the experiment. Figure 10A is the welcome scene - same as in Experiment1. Figure 10B is the next scene that provides instructions to the subjects about the task. The next scene is the instruction quiz shown in Figure 10C that ensures that subjects have read the instructions. If they get the quiz wrong, they are sent back to the instructions page, and requested to re-read them. This is followed by five practice trials in the order shown in Figure 9A. As an example, the second practice trial is shown in Figure 10D. At the end of each trial, a "Great Job" (Figure 10E) or "Out of Time" (Figure 10F) pop-up window is displayed depending on whether the subject collected all the diamonds in the environment or ran out of time before doing so. Both these pop-ups also specify the ratio of the number of diamonds collected to the total diamonds embedded in the environment. This provides feedback to the subjects about their performance in the trial before they continue to the next trial. At the end of the practice trials, subjects are presented with another instruction quiz as shown in Figure 7G, to ensure that they understand the task. If they answer the instruction quiz incorrectly, they are requested to re-read the instructions and re-do the practice trials, and are sent back to the instructions scene in Figure 7B.

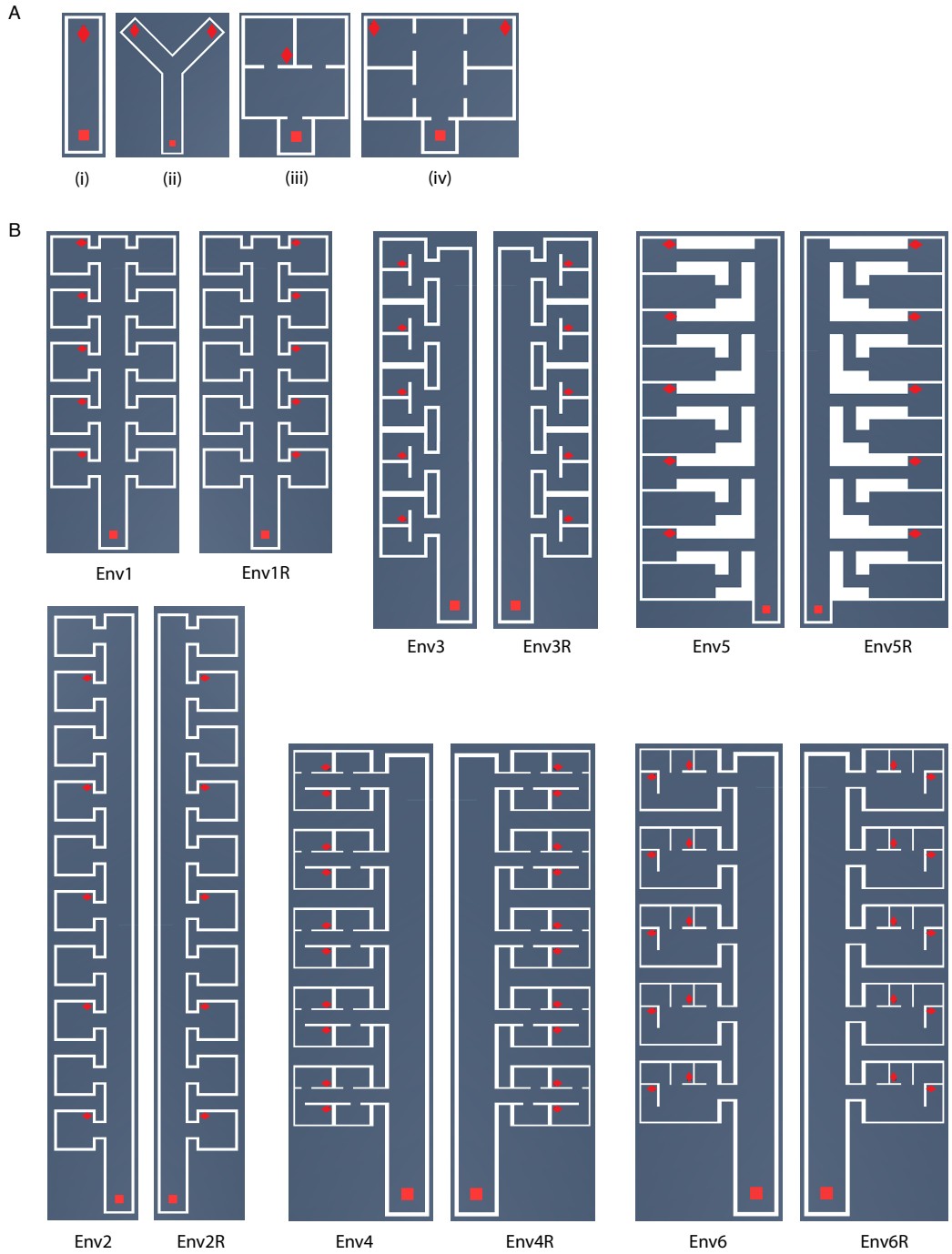

Figure 6: Top-down views of the stimuli used in Experiment1. **A.** Practice stimuli. **B.** Test stimuli. 'R' indicates reflected. The red square is a floor mat indicating the starting location. The red diamonds are the rewards.

If the subjects answer the instruction quiz correctly, they are shown additional instructions (Figure 10H) and another instruction quiz (Figure 10I) before moving onto the test trials. Next, the twelve test trials (see Figure 9B) are presented in succession. An example test trial is shown in Figure 10J. The test stimuli are grouped into two blocks containing six stimuli each, according to their difficulty level. For each subject, the order of stimulus presentation within blocks is randomized, while the blocks themselves are presented in a fixed order e.g., block1 (relatively simple environments - Env1,

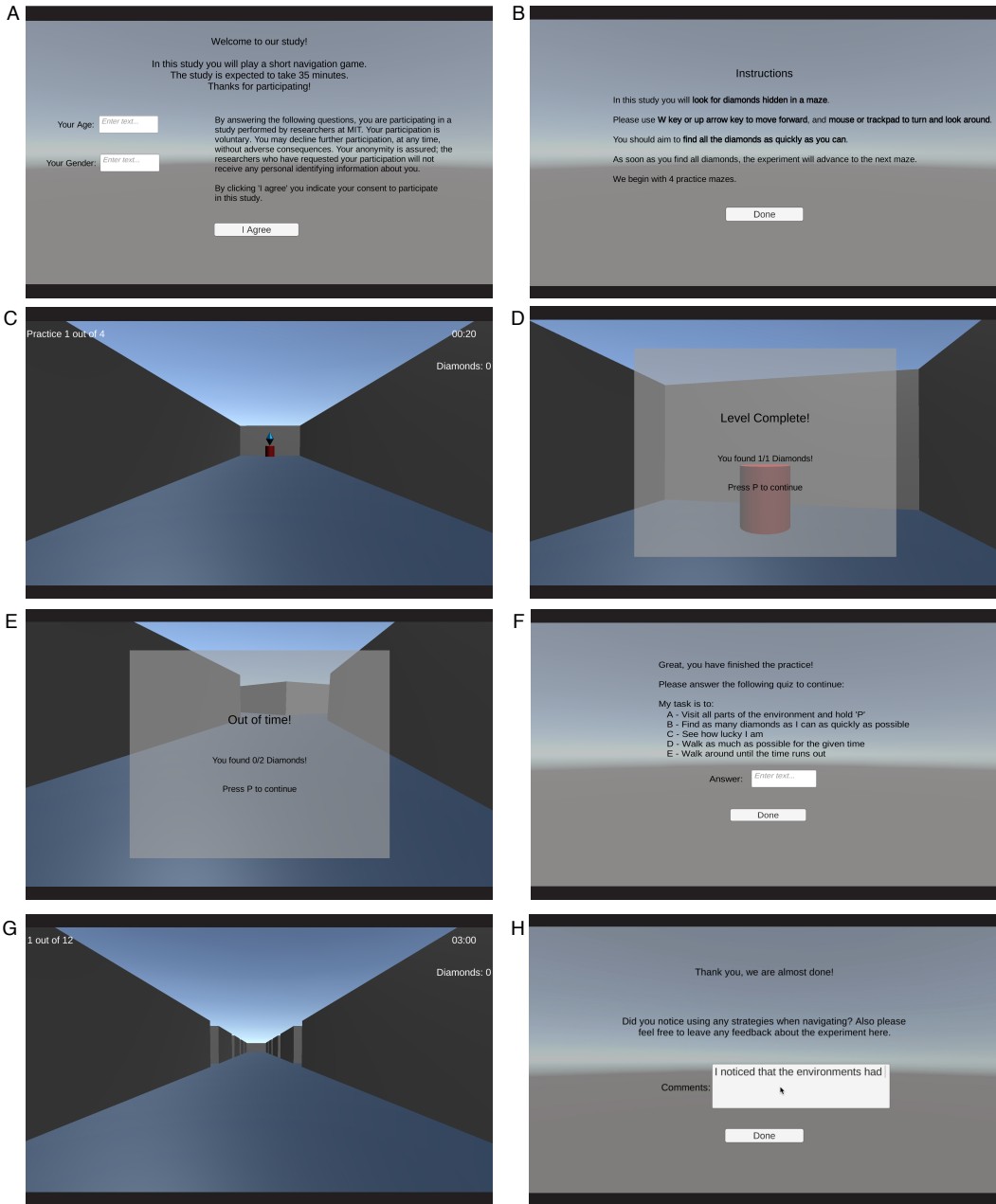

Figure 7: Screenshots from Experiment1. **A.** Welcome scene. **B.** Instructions. **C.** First practice trial. **D.** Pop-up window displayed at the end of a successful trial. **E.** Pop-up window displayed when the subject runs out of time before collecting all diamonds. **F.** Instruction Quiz. **G.** Sample first test trial. **H.** Concluding scene at the end.

Env1R, Env3, Env3R, Env5, Env5R) is presented first, followed by block2 (high complexity - Env2, Env2R, Env4, Env4R, Env6, Env6R).

After the test stimuli, subjects are subjected to a controls skill test unbeknownst to them, to test their skills using navigation controls. This test is introduced to ensure that subjects are able to navigate in WebUnity without undue difficulty, given the larger environments, and a relatively higher cognitive load in this experiment. The test is introduced at the end rather than at the beginning to account for the deterioration of control skills due to fatigue. Subjects are shown the scene in Figure 11A that tells them that they are about to enter an environment with ten diamonds, one in every room. They

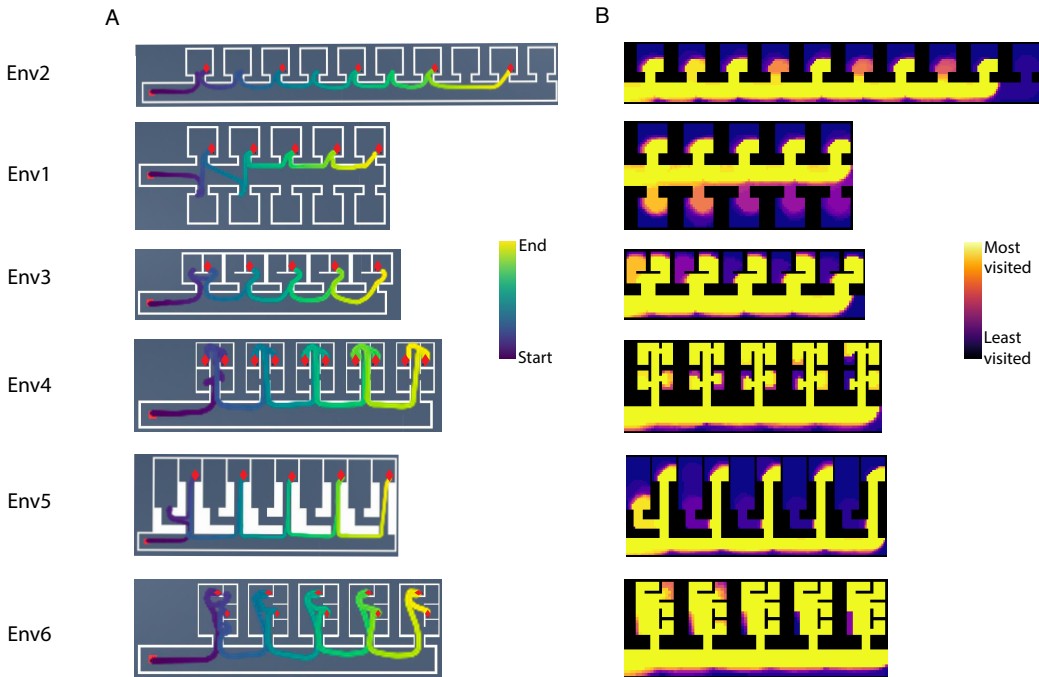

Figure 8: Trajectories and heat-maps from Experiment 1. **A**. Exploration trajectories of a representative subject. Stimuli are shown in the order presented to the subject (top - bottom). **B**. Visitation heat-maps aggregated across subjects. Each subject's visited area was computed in a 2D representation of the environment, using a circular radius of five grid cells centered at the cells traversed by the subject.

are instructed to visit every room to collect all the diamonds. Figure 11B shows the controls skill test trial with the top-down view of the stimulus shown in Figure 11C. To pass the test, subjects must collect all the diamonds in this trial. The test helps disqualify two types of subjects: i) subjects who do not complete the experiment i.e., wait at the entrance without exploring the spatial environments, in the hope of getting the base payment from the study; ii) subjects who have difficulty navigating in WebUnity due to inexperience/fatigue or difficulty with controls since they may not have the appropriately functioning control devices (mouse/keyboard). Subjects who failed the controls skill test were paid, but excluded from the analysis. Figure 11D shows the trajectory of a representative subject who failed the test. After the controls skill test trial, a final scene is shown to thank the subjects for their participation, and to get their comments/feedback.

Figure 12 shows the results from the experiment on the six base test stimuli (Env1-Env6). Panel A shows the trajectories of a representative subject. The stimuli are shown in the order presented to the subject (top - bottom). Panel B shows the heat-maps indicating the least to most visited regions in each of the base test environments, averaged across subjects. Separate heat-maps for when the base environments are shown during the first and the second presentation (mirrored with respect to the first presentation) are shown in Figure 14.

## A.3   MODEL LIKELIHOOD COMPARISONS

Our model likelihood analysis aims to compare the similarity of human decisions to model decisions. Specifically, we wanted to determine which models made the most human-like decisions. Because action-level similarity was too granular, we decided to compare room visitation similarity. We accomplished this by manually breaking each map into a set of convex rooms as shown in Figure 15. Using this map decomposition, we obtained an ordering of visited rooms for each human on each map. For each point on the human trajectory, we provided each model with the current human initial state and history of human observations. We queried the model to find an optimal plan from this state. The result of such a query is a policy tree that branches on observations and actions.

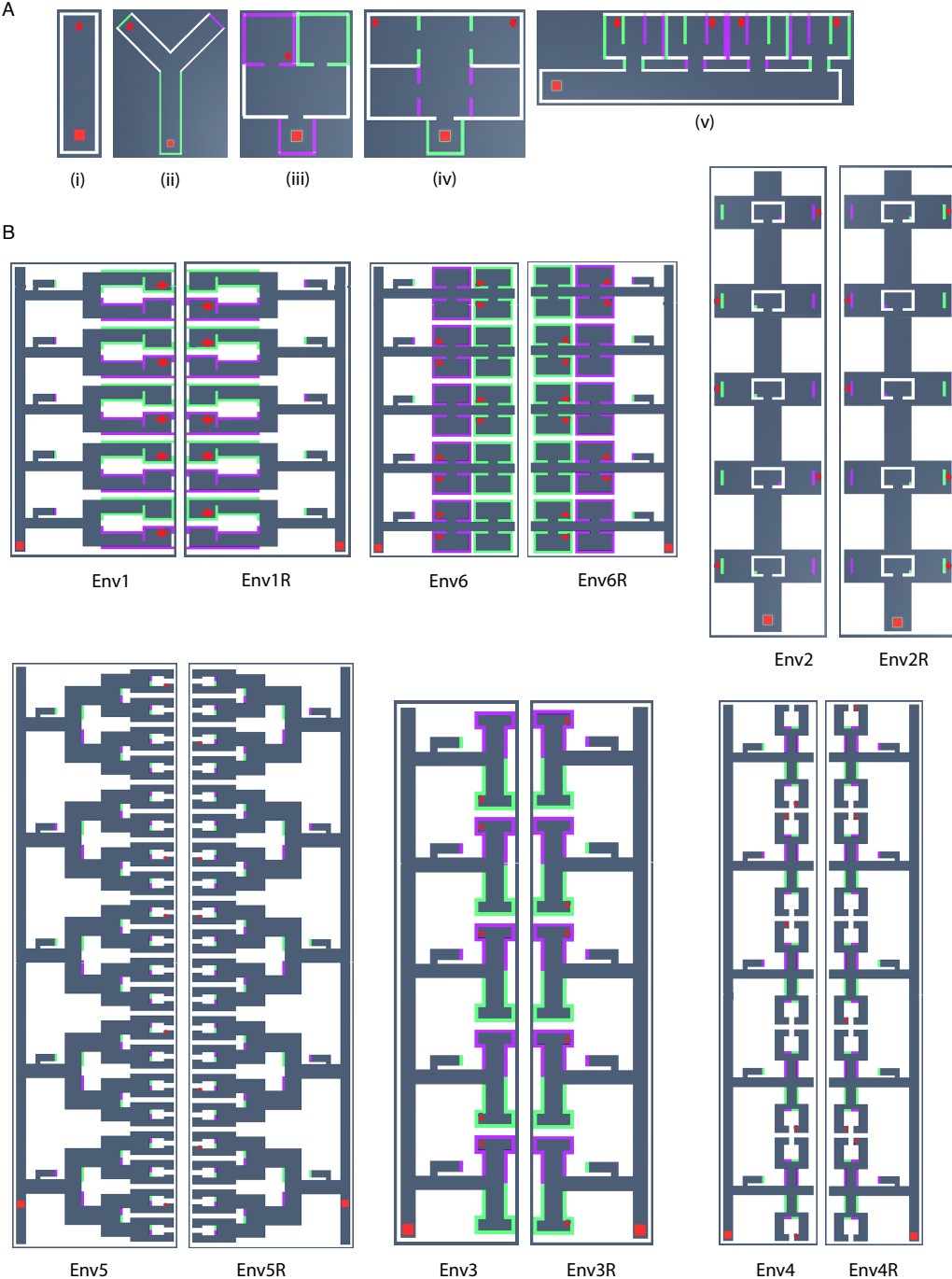

Figure 9: Top-down views of the stimuli used in Experiment 2. **A.** Practice stimuli. **B.** Test stimuli. 'R' indicates reflected. The red square is a floor mat indicating the starting location. The red diamonds are the rewards.

The most significant statistic of a POMDP policy tree is visitation frequency of nodes on the tree. To convert a policy tree to room visitation probabilities, we first flattened all the trajectories in the policy that led to a different room. We then summed and normalized their state visitation frequency in the policy tree. This process results in a distribution of rooms that the model is likely to visit given the human's experience. The model likelihood for a particular human is defined as the arithmetic mean of the log probabilities of the human's next room in the model's room visitation distribution.

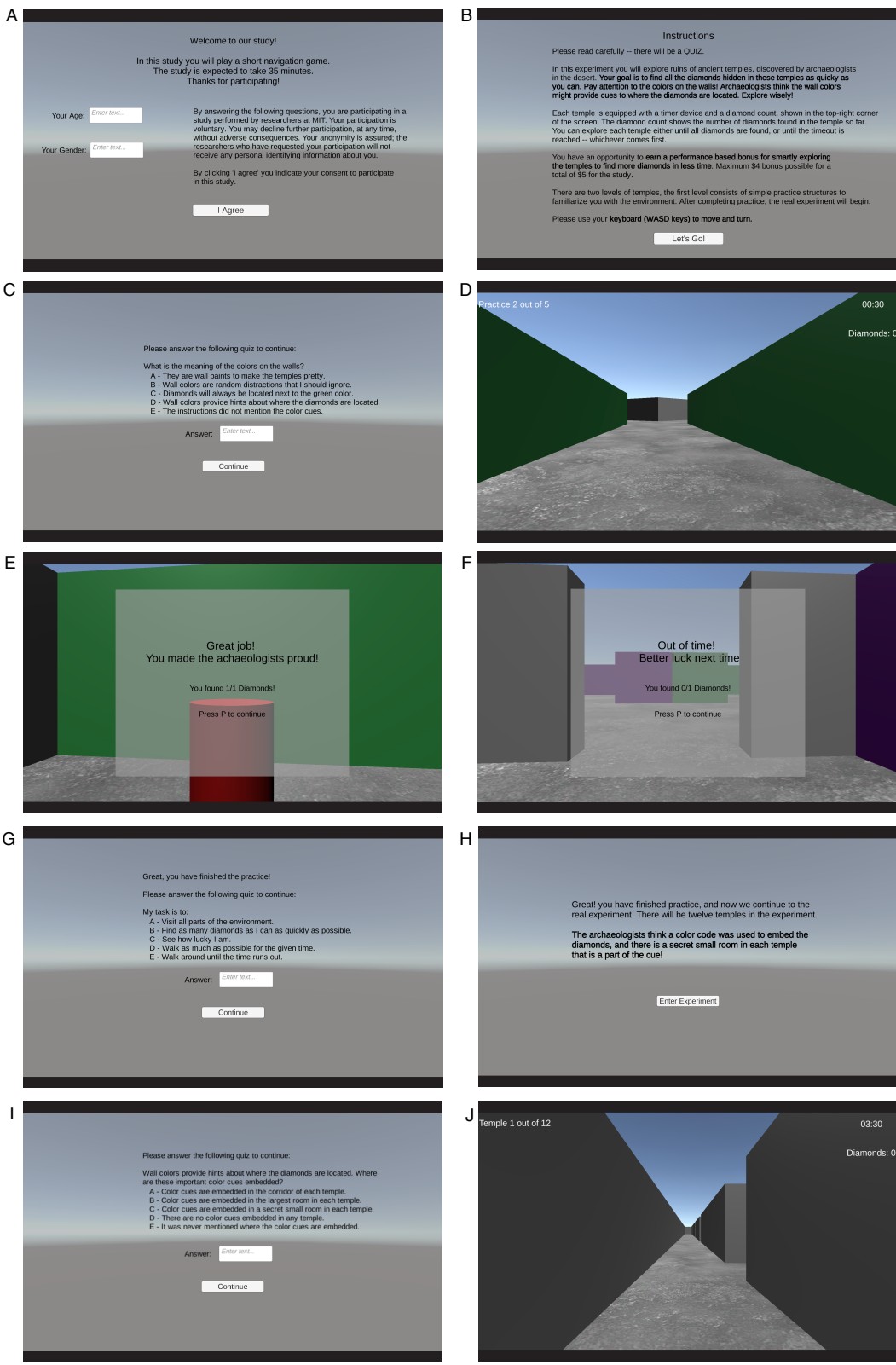

Figure 10: Screenshots from Experiment 2. **A.** Welcome scene. **B.** Instructions. **C.** Instruction Quiz 1. **D.** Second practice trial. **E.** Pop-up window displayed at the end of a successful trial. **F.** Pop-up window displayed when the subject runs out of time before collecting all diamonds. **G.** Instruction Quiz 2. **H.** Instructions for test trials. **I.** Instruction quiz 3. **J.** Sample first test trial.

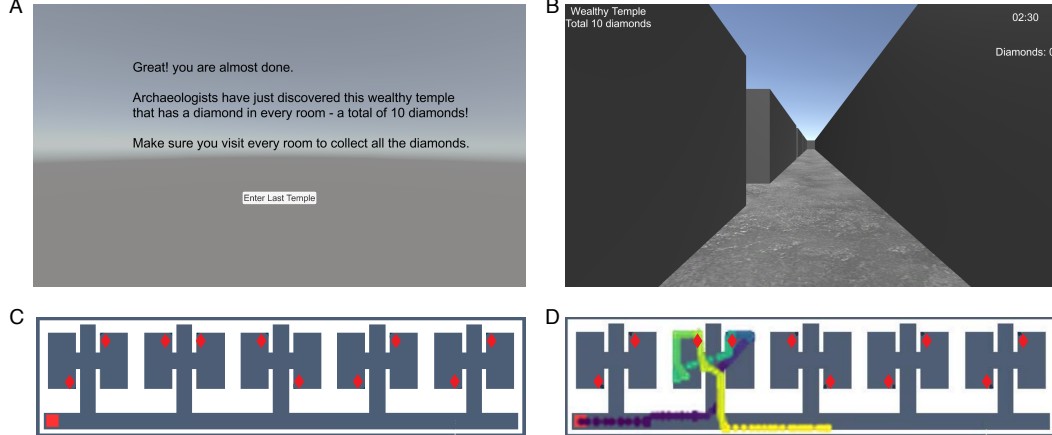

Figure 11: Controls skill test in Experiment 2. **A.** Instructions about the Controls skill trial. **B.** Controls skill trial. **C.** Top-down view of the stimulus used for the test. **D.** Exploration trajectory of a subject who failed the test.

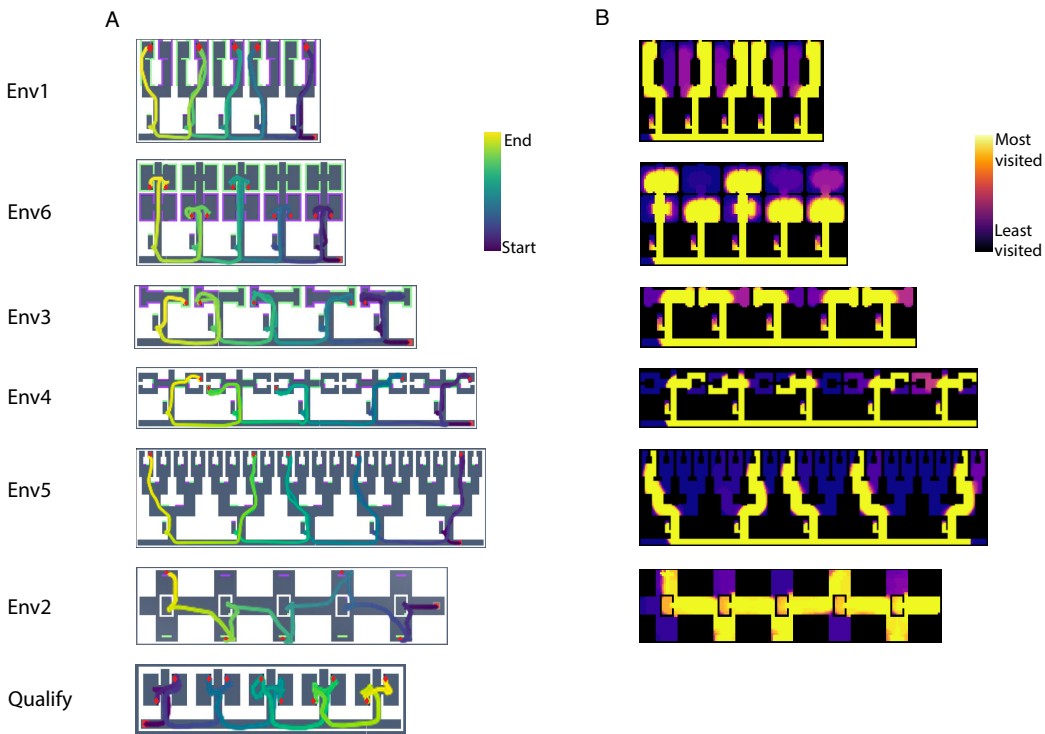

Figure 12: Trajectories and heat-maps from Experiment 2. **A.** Exploration trajectories of a single representative subject. Stimuli are shown in the order presented to the subject (top - bottom). **B.** Visitation heat-maps aggregated across subjects. Each subject's visited area was computed in a 2D representation of the environment, using a circular radius of five grid cells centered at the cells traversed by the subject.

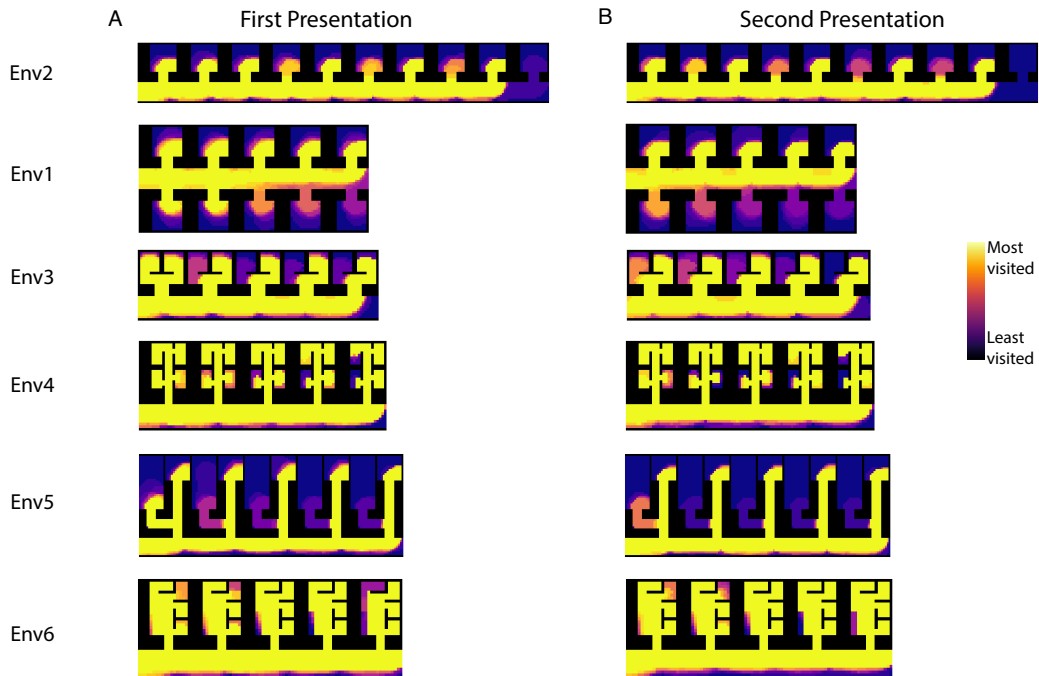

Figure 13: Experiment 1 visitation heat-maps aggregated across subjects. **A.** Heat-maps for the first presentation of environments. **B.** Heat-maps for the second presentation of environments. The stimuli in the second presentation were reflected versions of the ones in the first presentation as shown in Figure 6.

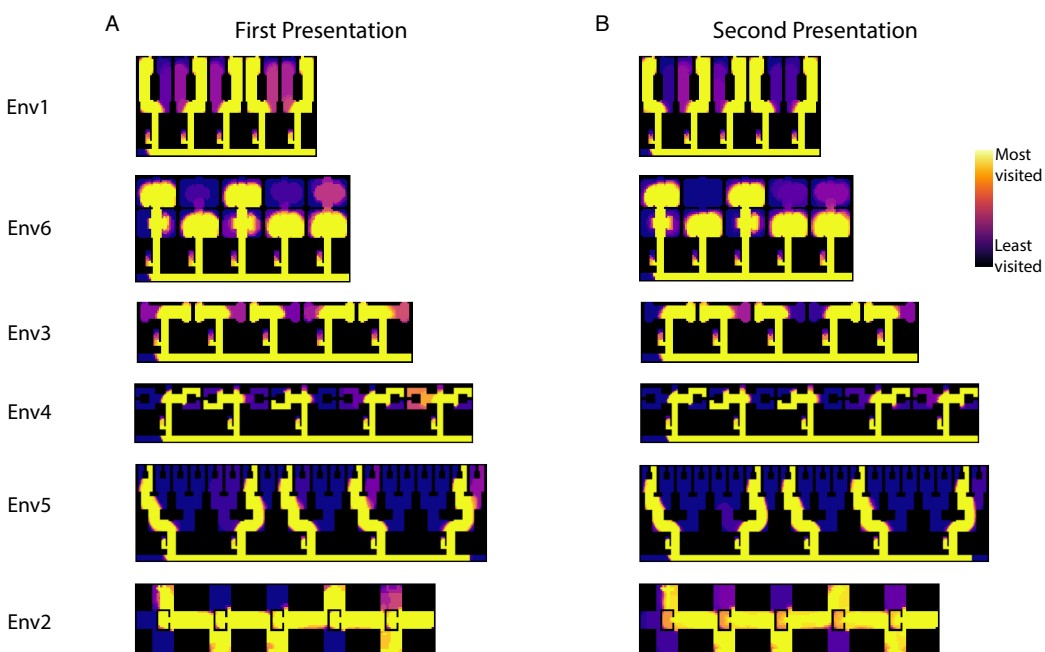

Figure 14: Experiment 2 visitation heat-maps aggregated across subjects. **A.** Heat-maps for when the base environments are shown during the first presentation. **B.** Heat-maps for when the base environments are shown during the second presentation. The stimuli in the second presentation were reflected versions of the ones in the first presentation as shown in Figure 9.

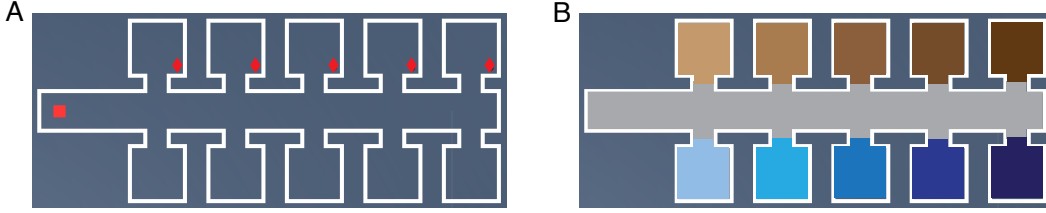

Figure 15: Illustration showing how the maps are qualitatively broken into convex rooms (discrete areas) **A.** Env1 stimulus from Experiment 1. **B.** Set of convex rooms defined for Env1. Each color indicates a separate convex room.

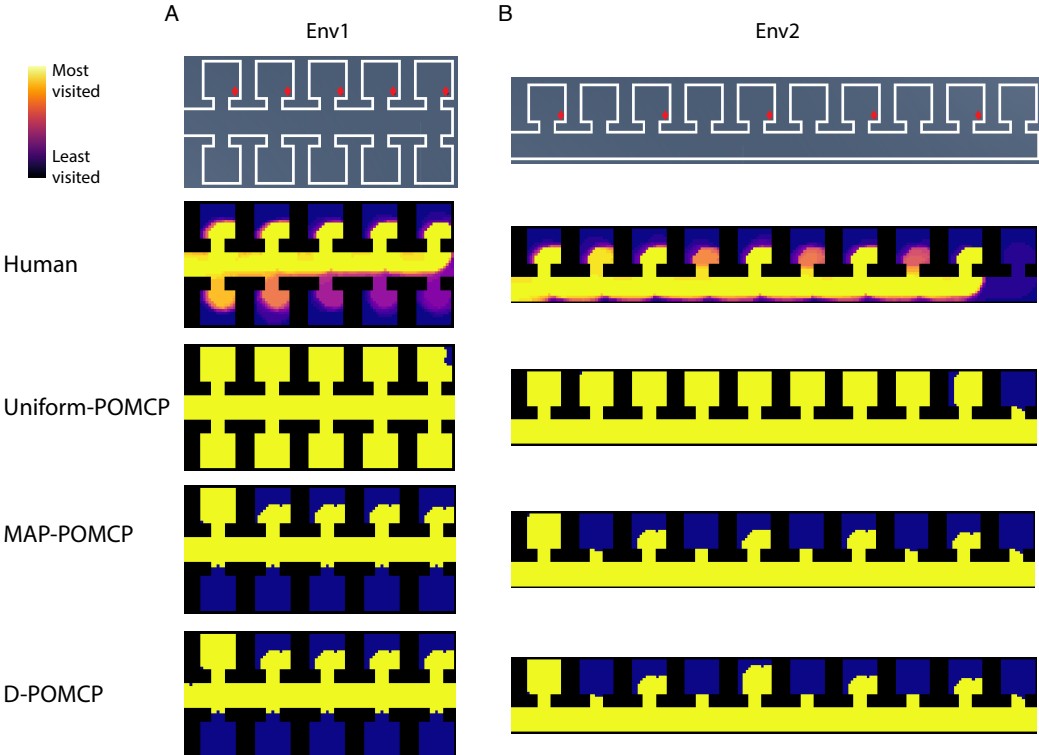

Figure 16: Model visitation heatmaps from two sample environments in Experiment 1. **A.** Env1. **B.** Env2. Top to bottom: top-down view of environment layouts; human visitation heatmaps aggregated across subjects and pairs of reflected environments; visitation heatmap from the Uniform-POMCP model; visitation heatmap from the MAP-POMCP model; visitation heatmap from the D-POMCP model. The visited area was computed in a 2D grid projection, using a circular radius of five grid cells around the agent.

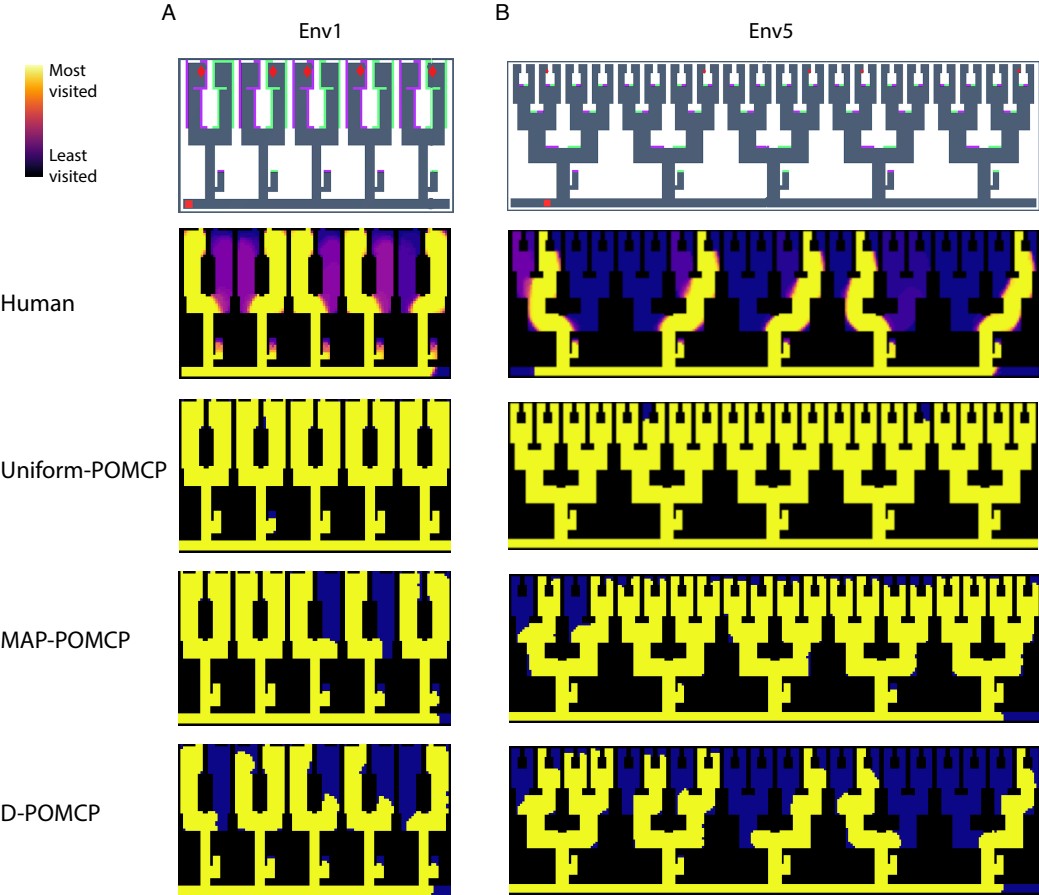

Figure 17: Model visitation heatmaps from two sample environments in Experiment 2. **A.** Env1. **B.** Env5. Top to bottom: top-down view of environment layouts; human visitation heatmaps aggregated across subjects; visitation heatmap from the Uniform-POMCP model; visitation heatmap from the MAP-POMCP model; visitation heatmap from the D-POMCP model. The visited area was computed in a 2D grid projection, using a circular radius of five grid cells around the agent.

