# OpenReview forum: "Map Induction: Compositional spatial submap learning for efficient exploration in novel environments"
_ICLR.cc/2022/Conference — ICLR 2022 Poster_

### Official Review · Reviewer_8xU7 · 2021-11-02

**Correctness:** 3
**Technical Novelty And Significance:** 4
**Empirical Novelty And Significance:** 4
**Recommendation:** 6
**Confidence:** 3

**Main Review:**

The article is well written and clear, providing a good motivation for the work and an good review of the related literature. I have on the other hand some concerns about the suitability of this paper for this conference and the methodology.

Although the author claim the dual goal of: "(1) to empirically study how humans learn spatial representations using a novel experimental Map Induction Task, and (2) describe a computational model of map induction that can be leveraged to optimize exploration in AI", point (2) is not really elaborated in the paper, nor does the paper provide experimental evidence to this claim.

Therefore, my first concern with this paper, is that this is effectively a Psychology article. There is nothing wrong with that of course, but the hypothesis, its relation to prior literature and methodology ought to be reviewed by an expert in the field.
For example, it is difficult for me to judge how well the results obtained on the proposed task would apply more broadly to human cognition in the real world. Although the literature review is generally good, a better discussion of how the paper's hypothesis fits with existing theories and previous experimental results would be useful.

The second concern is methodoligical. Although several hypotheses are mentioned in the text of the article, it is not clear how those hypotheses are supported from either the literature or experiments. For example:
- "We hypothesize that humans represent survey knowledge as composed of regions" [p. 3]
- "We hypothesize that while navigating novel environments, humans approximately infer a posterior distribution p(M|D) over its possible maps..." [p.5]
- "humans prefer map completions that are maximally descriptive of unseen portions of the environment" [p.6]
- "humans prefer map hypotheses that are minimally contradictory with previous observations." [p.6]
How confident are the authors that those assumptions hold, and what would be the consequence for their experimental results if they did not?

Moreover, the experimental results do not seem to clearly provide the evidence to support or discard the hypothesis. No hypothesis testing is provided. Concerning the number of participants in each experiment: Why using such a small number of participants if using Amazon mechanical Turk? Are the authors confident that this number is sufficient to prove/disprove the hypothesis?

**Summary Of The Paper:**

This paper investigates the concept of map induction for exploration in novel environments. This is an important problem for robotics applications, relevant for scaling up navigation of autonomous systems to large environments and for exploring unknown environments.

The article's main contribution is a new task and a set of experiments to evaluate the central hypothesis: "that humans use program induction to infer possible maps of unseen spaces, as made up of submaps encountered in the observed areas".

This is evaluated by proposing a novel map induction task that is the main contribution of the paper, as well as a set of probabilistic models that implement the different hypotheses considered in the paper.

**Summary Of The Review:**

In summary, the paper is generally well written and tackling and interesting question. The proposed task is interesting, but I am concerned that the paper does not provide a clear argument that the proposed methodology and analysis actually prove the hypothesis.
- This paper is mainly a Psychology study (as the computational model for AI part of the claims is not really demonstrated), and ought to be reviewed by an expert in experimental psychology.
- The analysis of the result does not provide clear evidence that the hypothesis is true
- The hypothesis should be framed more clearly with the existing literature, existing hypotheses of human spatial cognition and prior experimental evidence using established tasks.

---

> ### Author Response · Authors · 2021-11-22
> **Response to Reviewer 8xU7**
>
> We thank the reviewer for their support, and insightful thoughts. We appreciate your feedback and address your comments below:
>
> *"... point (2) is not really elaborated in the paper..."*
> - In the rebuttal, **we have edited the second goal stated in the last paragraph of section1.**
>
> - The updated version of goal 2 states the **potential** of the presented model and the conceptual framework to be used in AI, without overstating our contributions.
>
> - We have also **edited the first paragraph of the Discussion** to clarify our contributions. In the revised Discussion we now state that we address point two by showing that the performance of a Partially Observable Monte Carlo Planner can be improved by adding map induction.
> ***
> *"Therefore, my first concern with this paper, is that this is effectively a Psychology article...."*
> - We agree with the reviewer that the study of human cognition is indeed central to our work. However, we would like to argue that our work focusing on cognition is still consistent with the themes discussed in ICLR, given that we address the research question primarily through computational modelling. Many papers that take a similar approach have previously appeared in ICLR, for example:
>   -  [1] https://openreview.net/forum?id=--gvHfE3Xf5
>   -  [2] https://openreview.net/forum?id=w_7JMpGZRh0
>   -  [3] https://openreview.net/forum?id=Tp7kI90Htd
>
> - The papers previously presented in ICLR, deal with a wide range of cognitive and cognition-adjacent topics, such as transfer learning [1], social perception [2], visual processing [3].  Our work builds on this tradition - of thinking about cognition as computation through computational modelling -- and extends it by modelling efficient, cognitively-plausible map representations. The map induction model takes a step towards building artificial agents that perform human-like exploration, by showing that a POMCP planner performance can be improved by augmenting the planner with map induction.
> ***
> *"...  how the paper's hypothesis fits with existing theories and previous experimental results..."*
> - We have **added a paragraph at the end of section 3.2** to explain how the hypotheses outlined in this section fit with existing theories and previous experiments explained in Section1.
> ***
> *"The second concern is methodoligical. ..."*
> - We thank the reviewer for this question. Understanding the ways in which the framing of our hypothesis may be unclear to the readers is extremely helpful to us. Below we provide clarifications regarding the four examples provided by the reviewer.
>   - In the first point we formulate the map induction hypothesis verbally, using descriptive language: “humans represent survey knowledge as composed of regions”, introducing Map Induction in a concise form that is intuitive to a human reader. However, to rigorously test a cognitive hypothesis, we need to give a precise formal definition that can be implemented computationally for quantitative predictions. This is the standard way of presenting and evaluating cognitive hypotheses (e.g. this method is employed in [1-3]), which entails that the hypothetical phenomenon is first given descriptively in the Introduction, and then formally defined in subsequent sections specific to Computational Modelling, and Implementation.
>   - The second point refers to a formal definition of the map induction hypotheses, where we elaborate on what we mean by **map induction in formal and computational terms (Section 3)**.
>   - The assumptions in points three and four are formally expressed by the **likelihood equation (Section 3.1: map inference)**.
> - Thus all four assumptions are formally expressed by our model. We test the validity of these assumptions by conducting two human experiments. Experiment 1 is designed to establish whether humans use any form of map induction at all. Experiment 2 differentiates between two specific implementations of map induction.
>
> - **We have revised the manuscript (Section 1:  paragraphs 1, 6  and 7; Section 2: paragraphs 1 and 2, Section 3: paragraph 1, Section 3.1: all paragraphs)**  to clarify the framing of the map-induction hypothesis in descriptive, and formal terms.
> ***
> *"...Why using such a small number of participants if using Amazon mechanical Turk? ..."*
> - The number of participants needed for both experiments was established through extensive pilot experiments. **We have added a clarifying statement to the revised manuscript (Section 4, paragraph 1).**
>
> - More specifically, we have run at least three pilot experiments with the first version of the task (Experiment 1), and five pilot experiments with the second version (Experiment 2). The data collected in pilot experiments was statistically analysed to establish the number of subjects needed to achieve reasonable confidence intervals on the posterior model likelihoods, assuming the variance between subjects is stable between independent replications of the experimental procedure.

---

### Official Review · Reviewer_vNyW · 2021-11-02

**Correctness:** 3
**Technical Novelty And Significance:** 3
**Empirical Novelty And Significance:** 4
**Recommendation:** 8
**Confidence:** 4

**Main Review:**

Although it is a work focused on how humans plan and predict unseen space, it provides good inspiration for artificial intelligence. The work is novel, as program induction has not been applied to map prediction and spatial planning before. I believe it will also inspire much more in-depth experimental works.
One minor concern I have is about the experimental design and analysis: when participants visit mirrored environments, they may recognize it is an mirror image of a previously learned environment, and thus the predicted distribution of map may be different from when they first see that environment. It was not clear to me whether mirroring of a previously learned environment is part of the program induction, as I suppose most of the modeling is performed within game instead of across game. Please clarify this.
Further, due to the above reason, it seems less justified to combine the result of first visit of an environment and the visit of its mirror environment. It would be good to plot heatmaps for them separately in the appendix.

An illustration of how the authors manually breaking each map into a set of convex rooms is necessary for understanding the model fitting. Is the model likelihood geometric or arithmetic average of the predicted probability of humans' next room visitation across all convex rooms (last sentence of A.3)?


It was not super intuitive for me why D-POMCP would predict better than MAP-POMCP in experiment 2. Is it mainly because the reward locations themselves cannot be deterministically predicted as in Experiment 1?
Is there any component of softmax in the model's policy for choosing routes? If not, where does stochasticity come from in the predicted distribution of visitation by the models? If yes, will D-POMCP still win when the softmax temperature or epsilon-greedy are free parameters?

Are there any free parameters being tuned to fit data? If yes, what are they?

**Summary Of The Paper:**

This paper tests a hypothesis of how humans choose the path to explore and maximize reward in unvisited space based on their past experience. The central hypothesis is that human uses program induction to generate prediction of possible spatial map. Based on human behavior in two experiments, the paper further compared different models of map prediction, and demonstrated that humans actually consider the distribution of possible maps instead of only consider the most likely map.



**Summary Of The Review:**

The paper appears to be generally interesting and novel. Perhaps because of the page limit, not enough details that would be expected for a human behavioral modeling are provided, and I hope they can be added during revision.

---

> ### Author Response · Authors · 2021-11-22
> **Response to Reviewer vNyW**
>
> Thank you for the encouraging comments, and valuable suggestions. We appreciate your feedback and address your comments below:
>
> *"...It would be good to plot heatmaps for them separately in the appendix..."*
>
> - You are exactly right. Analysis of the behavioral data in Experiment 1 revealed that humans recognize that an environment is a mirror image of a previously learned environment. However, they may still explore the first unit of the mirrored environment exhaustively since they are not sure whether the reward distribution is the same in the mirrored version. **We have added Figure 13** to show the separately plotted heat-maps for first and second (mirrored) presentation of stimuli in Experiment1.
> - Although most of the modelling presented in this paper applies map induction within an environment, we were able to apply the same model across environments. This was done by maintaining a long term memory of the submap proposals and using them to generate map hypotheses before generating new proposals in any new environment. In this case new proposals are only generated if no valid map hypotheses can be generated from already existing proposals, following the principle of “recognition before generation”.
> - In Experiment 2  **(as explained in appendix section A.2)**, relative to the base test environments (Env1-Env6), the reflected versions (Env1R - Env6R) have a reflected geometry, however unlike Experiment 1, they have a different reward distribution. Thus the heatmaps were not combined across environments in this case. However,  **we have added Figure 14** to show the separately plotted heat-maps for when the base (original non-mirrored versions shown in Figure 9)  environments were shown during the first presentation and second presentation (mirrored with respect to the first presentation).
>
> ***
> *"An illustration of how the authors manually breaking each map into a set of convex rooms is necessary...."*
>
> -  **We have added Figure 15** for an illustration showing how the stimuli maps are broken into a set of convex rooms. This is a process of qualitatively defining rooms (discrete areas) for each stimuli, and is used only for the behavioral analysis.
> - The model likelihood is the arithmetic average of the predicted probability of humans’ next room visitation. **We have clarified this in the Appendix section A.3**.
>
> ***
> *"It was not super intuitive for me why D-POMCP would predict better than MAP-POMCP in experiment...."*
>
> - Yes, the reason why D-POMCP predicts better than the MAP-POMCP in experiment 2 is because the way the rewards are distributed in each unit is different (in contrast to experiment 1 where the reward locations are the same across units). Since the likelihood function takes the reward distribution into account, picking the most likely map from the posterior distribution would lead to erroneous predictions in this case.
>
> - The model is stochastic since it has a stochastic transition function. Instead of doing a softmax over a number of actions, it does a softmax over outcomes in the search tree (used during the Monte Carlo Tree Search). For instance, the model doesn’t know if walking forward into an unseen region leads to a wall or a reward. This stochasticity in the model leads it to assign different values to different action sequences. We are looking at the aggregate value of action sequences that are similar to human action sequences in comparison to the aggregate value of action sequences that are different, where similarity is determined by room visitation.
>
> ***
> *"Are there any free parameters being tuned to fit data? ...*
> - Although the parameters of the model were tuned to optimize model performance on the task, this was done independently of the human behavioral data. Thus, there are no free parameters that were tuned to fit the behavioral data collected through experiments.
> - The parameters used in our model are listed below. (They are described at: https://anonymous.4open.science/r/Map-Induction/model/spatial_planning/README.md).
>
>   -  --task-name: The map sequence to use as the environment for the agent e.g., test1 to refer to Env1 in Exp 1
>   - --agent-name:
>       - pomcp_simple: Uniform model
>       - pomcp_ssp: Distributional model
>       - pomcp_mle: MAP Model
>       - random_policy: Random actions
>       - landmark_policy: Naive landmark-based policy
>       - landmark_reward_policy: Another naive landmark-based policy
>   - --search-depth: 20
>   - --tree-queries: 500
>   - --discount-factor: 0.90
>   - --optimism: 1e-5
>   - --replan-strat: ‘every_step’

---

> > ### Comment · Reviewer_vNyW · 2021-11-29
> > **regarding arithmetic measn**
> >
> > Thanks a lot for replying to all the concerns. I highly appreciate it. I have only one suggestion: arithmetic mean is not a proper evaluation, because to perform model comparison, essentially you need to calculate p(data|model) or p(model|data). Comparing this quantity on the log scale is equivalent to comparing the geometric mean of the trial-wise (or action-wise) predictive probability of participants' choices given your model. I realize that I should have replied to your reply earlier before the ending of the discussion period. If possible, I would like to know the comparison result when you use geometric mean instead.

---

> > > ### Author Response · Authors · 2021-12-01
> > > **Response to follow-up question by Reviewer vNyW**
> > >
> > > Thank you for your question. We performed model comparison using log-likelihoods, and their means. The resulting plot is available at https://anonymous.4open.science/r/Map-Induction/model/illustrations/log_likelihoods.pdf.
> > >
> > > We find that the trends D-POMCP >= MAP-POMCP > Uniform-POMCP remain consistent when we use log-likelihoods. We will report log-likelihoods in the future.

---

> > > > ### Comment · Reviewer_vNyW · 2021-12-02
> > > > **response to reply**
> > > >
> > > > I am glad to see the results plotted as log likelihoods, and that at least both models are better than uniform-POMCP model. I suggest you update your conclusion in the final version based on the result displayed here. Another way to plot (as a suggestion) is to provide error bars showing the relative difference of each model from the best model (or the worst model) in **cross-validated total** log likelihood. The difference would be clearer when you use the total log likelihood (summing over all trials) - I suppose the numbers you just plotted were average trial-wise log likelihood. And that cross-validation will take care of any potential overfitting due to unequal numbers of parameters.

---

### Official Review · Reviewer_M48j · 2021-11-02

**Correctness:** 3
**Technical Novelty And Significance:** 2
**Empirical Novelty And Significance:** 3
**Recommendation:** 6
**Confidence:** 4

**Main Review:**

## Strengths

- The experiments are tailored to the main hypothesis of the paper and manage to highlight the argued points well. It is evident that exploration is more efficient under the assumed computational model when solving the reward collection task, which indicates that the map induction is indeed informing the planner.
- The conducted real-life study also seems well-constructed. The results in fig. 4, d) and fig. 5, d) support the idea that a fully-probabilistic map induction model comes closer to human behavior than an uninformed / a point-estimated (MAP) one.
- The presented figures are legible and convey the message of the paper well.

## Weaknesses

- The considered environments are very simplified and exhibit very regular structures (repeating patterns). This is understandable, they are tailored to the proposed hypothesis and highlight the arguments made in the paper, they are also useful for comparing human behavior to the computational model. However, I cannot judge how well the presented arguments would extrapolate to more realistic (natural) environments, where e.g. repetition patterns & regularities are much less obvious.
- I find the claim in the abstract that the proposed approach outperforms state-of-the-art planning algorithms to be overstated (largely because of the previous point).
- Since everything is discrete and because of the combinatorial nature of map induction, the cardinality of the space over possible maps will quickly explode. This is acknowledged in the conclusion, but I am not sure if a discretized approach like program induction will scale well, even if shorter programs are considered (it can hurt planning performance). Have you considered some form of continuous relaxations?
- I think the section on the assumed computational model can be improved, right now it lacks a rigorous specification of the assumed joint distribution / probabilistic model of all variables in question (e.g. how the factorization looks, initial prior on the map, etc.).
- In section 3.1.4, the POMDP is introduced with a state-space that captures both agent states & map hypotheses. In that context, I find the verbal description of the assumed transition model confusing: it implies normalizing the map hypothesis distribution after a movement action is executed. Can you clarify why this is necessary? I think the clarity of this section would be better if the assumed transition model is explicitly specified (in terms of a conditional distribution).
- Is the likelihood in section 3.1.3. a well-defined distribution? It needs to integrate to 1 over the space of possible observations D (which should be discrete), but based on the description it seems like there can be many observations for which it evaluates to 1 (e.g. if $\beta$ and $\gamma$ are 0).
- Ideally, I would like to see visitation heatmaps from the computational model next to the ones in fig. 4 and fig. 5 from the real-life study, to judge whether the visitation patterns are similar / where they differ. The aggregated results in fig 4. c) d) and fig 5. c) d) somewhat obfuscate the comparison.


**Summary Of The Paper:**

This work is motivated by the hypothesis that humans maintain a hierarchical spatial representation when exploring new environments, such that shared patterns (due to the hierarchy) between spatial regions can be used to predict how still unvisited places would look. A simple discrete 2D computational model based on probabilistic program induction is proposed for predicting the map at yet unseen places. The model assumes a discrete set of possible transformations (flips, rotations, concatenations, etc.) of small extracted regions (submaps of previous observations), based on which a distribution over possible completions of empty map regions can be formulated.

A control task is then considered, in which an agent needs to explore a completely new environment and collect rewards (tokens) in the process. Under strong structural assumptions for the test environments (e.g. it is ensured that patterns actually repeat in a very uniform way, rooms look the same, rewards are placed consistently, etc.), it is shown that model-based planning under the map-induction model results in more efficient exploration & reward-collection behavior. Furthermore, a study with actual human subjects is conducted where they need to solve the same task. Results suggest that real-life exploration behavior is consistent with the most expressive version of the proposed computational model, where a full distribution over possible map completions is maintained (as opposed to a MAP estimate or an uninformed model).


**Summary Of The Review:**

I find the idea of the paper interesting, and I think the selected suite of experiments highlights the points the paper makes rather well. However, I cannot tell if the insights about map induction would extrapolate to more realistic cases, as the considered environments are very regular & specific in their repeating patterns, applicability might be limited. I also think the clarity of the model specification & the assumed probabilistic assumptions can be improved, I listed my main concerns in the main review.

---

> ### Author Response · Authors · 2021-11-22
> **Response to Reviewer M48j**
>
> Thank you for your thoughtful comments and suggestions. We appreciate your feedback and address your comments below:
>
> *".. I cannot judge how well the presented arguments would extrapolate to more realistic (natural) environments, where e.g. repetition patterns & regularities are much less obvious..."*
>
> - The arguments presented in the paper would naturally extrapolate to realistic environments that have regularities e.g., multistory apartment buildings, hospitals, hotels, food courts, parking areas etc. However, in environments where regularities are less obvious, the models presented in this paper would still induce the map layout based on prior observations and replan when the induced map deviates from new observations (more likely in parts of the environment that are not predictable). In this case, there may be small regions of uncertainty in the induced map, where either a uniform distribution can be assumed, or more suitably they can be approximately inferred based on the principle of contiguity i.e., contiguous or nearby regions are likely to be similar.  This is supported by previous work **(discussed in Section 1: second paragraph)** that suggests that human spatial representations are approximate, and that instead of learning a global metric map, humans learn spatial representations by combining redundant observations of local regions.
>
> - However, we understand that more comprehensive generative models might be needed for map induction in complex environments, in order to optimally induce the generative programs used to generate these environments. This is an active area for future work that we are pursuing. However, designing human behavioral experiments is more challenging in these complex environments because of the longer time-course of learning (as explained in the common comment above). **This is something we are actively thinking about for future work.  We have edited the third paragraph in Section 5 to reflect this intent.**
>
> ***
>
> *"I find the claim in the abstract that the proposed approach outperforms state-of-the-art planning algorithms to be overstated..."*
>
> - Thank you for expressing your concern. We have **reworded the abstract to ensure that it reflects our work, and doesn’t overstate it**.
> ***
>
> *"... because of the combinatorial nature of map induction, the cardinality of the space over possible maps will quickly explode. ...."*
>
> - Instead of enumerating all possible map completions, in future work we plan to use Markov Chain Monte Carlo Sampling to sample only the most promising map completions for scalability. **We mention this in a footnote in section 3.1: paragraph 4, on page 6.**
> - The main thesis of this paper is that compositionality of spatial primitives is an important aspect of map induction. Eventually we also want to extend this work to using continuous high dimensional primitives rather than the discretized primitives used in the current work, by implementing planning in a partially observable continuous domain.
>
> ***
> *"I think the section on the assumed computational model can be improved ..."*
>
> - We have **updated section 3.1 (especially paragraph 4: Map Inference)** to clarify how the posterior distribution, likelihood and initial priors on the maps are computed.
> - For simplicity, in this paper we assume a uniform distribution over all the enumerated map hypotheses generated by the Map Generator. In the current implementation, the space of possible hypotheses is small, so this is feasible, however in theory, the model architecture also supports a more realistic assignment of prior probability distribution such that the probabilities are derived from the generative grammar. This allows the specification of constraints e.g., for making shorter programs as well as regions that were recently used for successful prediction more likely.
>
> ***
> *"...I find the verbal description of the assumed transition model confusing..."*
>
> - Apologies for the confusing wording. There is no normalization of the map hypothesis after a movement action is executed. We have edited Section 3.1.1 (previously section 3.1.4) to correct this and to make it clearer.
> ***
> *"Is the likelihood in section 3.1.3. a well-defined distribution? ..."*
>
> - There was a writing error, and we apologize for that. We have **edited the likelihood equation in section 3.1 to reflect proportionality rather than equality**.
> - The likelihood function is a density function over the space of possible observations D and integrates to 1 over that space. However, the likelihood function doesn’t integrate to 1 over the space of maps M. This is because the likelihood is the probability of observing data D, given the induced map, and there is no reason for the integral of likelihood over all maps to sum to 1.
>
> ***
> *".. visitation heatmaps from the computational model"...*
>
> - **We have added Figure 16** : visitation heat-maps from the three hypothesized computational models for two sample environments from Experiment 1.

---

### Official Review · Reviewer_toMF · 2021-11-04

**Correctness:** 3
**Technical Novelty And Significance:** 4
**Empirical Novelty And Significance:** 2
**Recommendation:** 6
**Confidence:** 3

**Main Review:**

**Strengths:**

- While there are multiple heuristic / learnt models for exploration of novel spaces, modeling how humans explore novel environments is an interesting area of research. In this paper, the authors present an interesting take on this problem by treating it as program induction — inferring the program that will lead to a correct map of the environment based on past observations.
- The experimental setup and inferences made through the experiments were easy to grasp and made sense to me. The experiments were designed to test the two main hypothesis (1. Does humans rely on some sort of map induction for exploration as opposed to uniformly visiting all the map regions? 2) Do humans seek information about the task to score different possible map completion hypotheses?)

**Weaknesses:**

- A major concern for this paper is that the task was designed keeping the hypothesis is mind. For instance, the maps and 3D environments are synthetically constructed such that they have repeated blocks. Given that the environments were created to have repeatability, humans performing the task will obviously pickup on that signal. It would have been better if the experiment was conducted using publicly available layouts (3D scans like Gibson, Matterport etc).
- On a related note, the environments are composed of simple units and are almost grid-world like. For such environments, production rules (Table 1) used to generate map hypotheses are simple. Can the authors comment on how this will generalize to generic environments which might have fairly arbitrary layouts?

**Updates after Rebuttal**
Thank you authors for your responses and clarifications to my concerns. I think the hypotheses presented in the paper are interesting and are tested appropriately. Even though the paper lacks experiments on more complicated environments such as Gibson, Matterport, I think the ideas presented in the paper are good first steps and might be useful to the broader community (including myself). Thus, I am happy to update my scores.

**Summary Of The Paper:**

In this paper, the authors try to model how humans explore novel environments. The main hypothesis tested in the paper is that humans think of maps of unseen places as compositions of submaps in observed areas. Towards this, the authors propose a new "Map Induction Task" (MIT) to study how humans explore novel maps. Additionally, they try to model the exploration behavior exhibited by humans as a hierarchical bayesian generative framework which generates a distribution over possible maps given past observations, and use this distribution to plan. They present results on their  task (MIT) which validate the aforementioned hypothesis

**Summary Of The Review:**

The authors present an interesting hypothesis regarding modeling human exploration behavior. However, I am not convinced that the experimental setup captures general human exploration. The experimental setup were synthetically created to have repeatability and doesn't capture the general layouts (of indoor spaces or otherwise). Thus, I am not convinced that these hypotheses will hold true when tested on realistic environments.

---

> ### Author Response · Authors · 2021-11-22
> **Response to Reviewer toMF**
>
> Thank you for the encouraging comments, stimulating questions and suggestions. We address your comments below:
>
> *"A major concern for this paper is that the task was designed keeping the hypothesis is mind."*
>
> - Thank you for pointing this out. We agree that we designed the tasks in order to differentiate between the stated hypotheses. Constructing experimental stimuli with the intention to differentiate between alternative hypotheses is a common approach in experimental psychology and cognitive science.
>
> - Specifically, the task in Experiment 1 was designed to test whether humans use *map induction* (as implemented by the MAP-POMCP and D-POMCP models) or use the naive exploration implemented by the Uniform-POMCP model. The environments with repeatability used in Experiment 1 were sufficient to test whether humans use map induction as a cognitive mechanism. Since we find that humans optimize exploration in these environments rather than exploring naively, what computational cognitive mechanisms they might be using to do so is a non-trivial question. The map induction models (with spatial priors) that we present as our hypotheses (MAP-POMCP and D-POMCP model) present possible solutions to this problem.
>
> - The environments used in Experiment 2 are composed of at least two non-identical repeated blocks with different ‘cue-rooms’ and reward distributions. These environments were sufficient to test whether human exploration is best explained by MAP-POMCP or D-POMCP models.
>
> ***
> *"It would have been better if the experiment was conducted using publicly available layouts (3D scans like Gibson, Matterport etc"*
>
> - We agree that using these more complex environments can help us answer the following question: Do humans use map induction in more complex environments with arbitrary layouts?
>
> - Given that we have shown that humans use map induction in regular environments that have a predictable structure, it seems likely that they would also be able to use map induction in other environments that have some predictability i.e., where some parts but not whole of the layout can be predicted. In such environments, the models presented in this paper would induce the layout based on prior observations and replan when the induced map deviates from new observations (more likely in parts of the environment that are not predictable).
>
> - We understand that more comprehensive generative models might be needed for map induction in complex environments like the Matterport dataset, in order to optimally induce the generative programs used to generate these environments. This is an active area for future work that we are pursuing. However, designing human behavioral experiments is more challenging in these complex environments because of the longer time-course of learning (as explained in the common comment above). **This is something we are actively thinking about for future work. We have edited the third paragraph in Section 5 to reflect this intent.**
>
> - For the purposes of the study in this paper, we sought to use the simplest dataset that was sufficient to differentiate between our hypotheses and could help us test whether humans use map induction during exploration at all. The dataset needed to (1) contain environments with a layout that can in theory be predicted from partial observations (e.g., apartment blocks, malls, city streets in a grid); and (2) be integrated into Unity, to enable human data collection. We appreciate the reviewer's suggestion, and we will keep it in mind as we consider publicly available datasets, which we could use in future work.
>
> ***
> *"..the environments are composed of simple units and are almost grid-world like..."*
>
> - In a non grid-like world the production rules in Table 1 can still be used to generate map hypotheses with some uncertainty. For instance, arbitrarily shaped spatial regions can still be transformed (reflected, rotated, concatenated), however since they can’t be perfectly tiled, there will be small regions of uncertainty in the induced map. This is supported by previous work **(discussed in Section 1: second paragraph)** that suggests that human spatial representations are approximate, and that instead of learning a global metric map, humans learn spatial representations by combining observations of local regions.
>
> - In these regions of uncertainty, either a uniform distribution can be assumed, or more suitably an additional production rule for extending the spatial region can be added based on the principle of contiguity i.e., contiguous or nearby regions are likely to be similar.

---

### Author Response · Authors · 2021-11-22
**Addressing a common point raised by the reviews**

We thank the reviewers for their encouraging comments, thoughtful questions, and suggestions. We will use this top level comment to emphasize the main contributions of our work and address a common point raised by reviews, namely the “highly regular structure of the experimental environments”, and a suggestion to “use more complex environments, where repetition is less obvious”. Following this general response, we will address reviewers individually. We request the reviewers to ask any follow up questions they may have without hesitation.

The main conceptual contribution of our work is the idea that agents build a distribution of possible maps through an inductive process after seeing a partial piece of the environment. This Map Induction process is a conceptually new way to think about how humans navigate through the world and our goal is to test whether humans might use such an internal algorithm as they navigate through spaces in few- and zero-shot settings. Note that in order to build the most simple and testable correspondence between models and human performance, and thus test the conceptual framework, we  have used simple environments. To our knowledge, this idea of inducing maps is novel and hasn't been proposed in the literature before.

We thank the reviewers for the suggestion to consider more complex environments. We agree that extending the study toward more complex environments -- such as layouts that combine several types of sub-regions, and environments in which only a part of the layout is predictable -- is an important next step. We are currently working on extending the map induction model to more complex maps of this kind.

One notable challenge with learning more complex or partially structured environments lies in the time-course of learning. The difficulty of the current human behavioral experiments was calibrated so that naive human subjects, unfamiliar with Unity or the experimental hypothesis, could reasonably learn the underlying maps in a 30 minute experimental session, while experiencing the environments from a first-person perspective. In real-life contexts people do learn more complex map induction rules (e.g. locating bus stops, food courts, and parking) but have a longer time-frame to do so. While the design of the human study, and the clarity of the experimental procedure are major considerations informing our current experiment design, we intend to extend the map induction model to more complex environments in future work.

---

### Author Response · Authors · 2021-11-22
**Thank you for your feedback! - changes in the rebuttal revision**

We thank all reviewers for their time and for their constructive feedback. It helped us a lot to improve our paper. The rebuttal revision of the paper incorporates your sound suggestions. Concretely, following are all the changes we made:

- Updated the third paragraph in Section 5 to reflect our intent to study map induction in larger, more naturalistic environments in future work.

- Reworded the abstract to ensure that it reflects our work, and doesn’t overstate (removed strong words like outperform).

- Updated the footnote in section 3.1: paragraph 4, on page 6 (indicating the methods that can be used for scaling our map induction framework to more complex environments)

- Updated section 3.1 (especially paragraph 4: Map Inference) to clarify how the posterior distribution, likelihood and initial priors on the maps are computed.

- Updated the likelihood equation in section 3.1 (map inference) to reflect proportionality rather than equality.

- Added Figure 13 and Figure 14 to the appendix to show heat-maps separately for first and second (mirrored relative to the first) presentation of stimuli.

- Added Figure 15 to illustrate how stimuli maps are broken into convex rooms and also edited the corresponding Appendix section A.3.

- Updated the second goal stated in the last paragraph of section1 to suggest the potential of the model for improving exploration in AI.

- Updated the first paragraph of the Discussion to clarify our contributions.

- Added a paragraph at the end of section 3.2 to explain how the hypotheses outlined in this section fit with existing theories and previous experiments explained in Section1.

- Revised the manuscript (Section 1: paragraphs 1, 6 and 7; Section 2: paragraphs 1 and 2, Section 3: paragraph 1, Section 3.1: all paragraphs) to clarify the framing of the map-induction hypothesis in descriptive, and formal terms.

- Added a clarifying statement to the revised manuscript (Section 4, paragraph 1) on how the number of participants for each experiment were determined.

- Added Figure 16 showing the visitation heat-maps from the three hypothesized computational models for two sample environments from Experiment 1.

---

### Decision · Program_Chairs · 2022-01-20

**Decision:**

Accept (Poster)

**Comment:**

This paper presents a hierarchical Bayesian approach to exploration in grid worlds.  The paper considers the hypothesis that humans maintain a hierarchical representation when exploring a space, where the distribution over unknown space can be modeled with a structured probabilistic program.  The paper compares the behavior of people during exploration tasks to the behavior of a Bayesian model under different distributional approximations.  The results indicate that people can behave similarly to a sophisticated Bayesian model on small grid world domains.

The reviews highlighted several concerns about the paper.  One initial concern was that the experimental domain is too simple and small compared to real world environments encountered by robots or humans.  However, this work is similar in scope to other exploration work in reinforcement learning and psychology studies, where tiny grid worlds are still commonly used to gather insight.  Thus, this concern does not reduce the potential contribution of the paper.  The reviewers raised several other concerns about the work that were largely addressed by the author response.  The remaining reviewer concerns centered on the limited strength of the evidence in the experiments, but the reviewers expect the paper will still be of interest to the broader research community.

Four reviewers indicate to accept this paper for its contribution of a study into the use of probabilistic program induction to infer possible completions of maps in small environments.  The paper is therefore accepted.